



# Raman spectroscopy in thrust-stacked carbonates: an investigation of spectral parameters with implications for temperature calculations in strained samples

Lauren Kedar[1], Clare E. Bond[1], David K. Muirhead[1]

5 [1]University of Aberdeen, School of Geoscience, Aberdeen, AB24 3UE, UK

*Correspondence to*: Lauren Kedar (l.kedar@abdn.ac.uk)

**Abstract.** Raman spectroscopy is commonly used to estimate peak temperatures in rocks containing organic carbon. In geological settings such as fold-thrust belts, temperature constraints are particularly important as complex burial and exhumation histories cannot easily be modelled. Many authors have developed equations to determine peak tempertaures from 10 Raman spectral parameters, most recently to temperatures as low as 75°C. However, recent work has shown that Raman spectra can be affected by strain as well as temperature. Fold-thrust systems are often highly deformed on multiple scales, with deformation characterised by faults and shear zones, and therefore temperatures derived from Raman spectra in these settings may be erroneous. In this study, we investigate how the four most common Raman spectral parameters and ratios change through a thrust-stacked carbonate sequence. By comparing samples from relatively low-strain localities to those on thrust 15 planes and in shear zones, we show maximum differences of 0.16 for I[d]/I[g] and 0.11 for R2, while FWHM[d] and Raman Band Separation show no significant change between low and high strained samples. Plausible frictional heating temperatures of faulted samples suggest that the observed changes in Raman spectra are not the result of frictional heating. We apply three equations used to derive the peak temperatures from Raman spectra to our data to investigate the implications on predicted temperatures between strained and unstrained samples. All three equations produce different temperature gradients with depth 20 in unstrained samples. We observe that individual equations exhibit apparently varying sensitivities to strain, but calculated temperatures can be upto 140°C different for adjacent strained and unstrained samples using the same temperature equation. These results have implications for how temperatures are determined in strained rock samples from Raman spectra.

## 1 Introduction

Raman spectroscopy can be used to provide information on the nanostructure of organic carbon in rocks. Since this 25 nanostructure changes irreversibly with increasing temperature, Raman is a useful tool for establishing the peak temperature of rocks in a variety of settings. In turn, peak temperatures can provide information about sedimentary burial conditions (e.g. Ferrari and Robertson, 2001; Beyssac et al., 2002; Muirhead et al., 2012; 2017a; Schito et al., 2017), low-grade contact metamorphism (e.g. Aoya et al., 2010; Chen et al., 2017; Muirhead et al., 2017b) and tectonic thrust stacking. (Nibourel et al., 2018; Muirhead et al., 2019). Much work has been done to develop reliable equations which can be used to calculate





temperatures from Raman spectral parameters across a range of settings and geological processes (e.g. Beyssac et al., 2002;
Lahfid et al., 2010; Kouketsu et al., 2014; Schito and Corrado, 2018; Wilkins et al., 2018). With increasing understanding of
organic carbon nanostructure, such equations have been recently applied to a much wider range of temperatures (down to
75°C; Schito and Corrado, 2018; Muirhead et al., 2019) and geological settings such as fold-thrust belts (Nibourel et al., 2018,
2021; Muirhead et al., 2019).

Fold-thrust systems, driven by deformation, are subject to complex burial and exhumation histories and so the temperature
history for a specific rock within a thrust stacked sequence is often not straightforward. In addition to this thermal complexity,
recent work has shown Raman spectra to be affected by strain (Kwiecinska et al., 2010; Kitamura et al., 2012; Furuichi et al.,
2015; Kedar et al., 2020). Therefore, if temperatures are to be investigated using Raman spectroscopy in strained terrains, the
individual effects of strain and temperature on Raman spectra need to be isolated from one another.

In this study, we analyse the impact of strain and temperature on organic carbon nanostructure in a fold-thrust belt. Raman
spectral parameters for a suite of samples, taken from a transect across a thrust-stacked carbonate sequence in the French Alps,
are plotted on a cross-section. We use this visualisation, and the associated Raman data, to investigate how the most commonly-
used Raman spectral parameters in published temperature equations – peak intensity ratio (I[d]/I[g]), Raman band separation
(RBS), peak width (FWHM), and peak area ratio (R2) – change through the sequence. Using three Raman-based temperature

equations: Lahfid et al. (2010), Kouketsu et al. (2014) and Schito and Corrado (2018), we investigate how these different
equations yield varying temperature predictions through the stratigraphy, and consider the implications for the predicted
thermal gradient. We also identify samples which are affected by locally high strain, such as thrust faults and shear zones, and
assess how the Raman spectra of these samples differ from adjacent unstrained samples. By applying temperature equations
(Lahfid et al., 2010; Kouketsu et al., 2014; Schito and Corrado, 2018) and by quantifying how each Raman spectral parameter

changes in strained samples, we assess the sensitivity to strain of each, and from this the impact on predicted temperatures.
We compare faulted samples to those from a ductile shear zone to investigate the effect of seismic slip events vs. aseismic
creep on Raman spectra, an important step towards separating the effects of strain and temperature. We discuss the implications
of our findings on the the relative merits and limitations of published temperature equations to predict geothermal gradients in
thrust stacked sequences  and the use of Raman spectra to predict temperatures in strained rock samples.

**2 Geological setting**

The Haut Giffre region of the French Alps (Fig. 1a, 1b) encompasses 3,000 m of Jurassic-Cretaceous carbonates, split into six
broad units (Fig. 1c) of contrasting mechanical properties. Each unit has a characteristic bed thickness, ranging from 0.005 m
in shale-rich layers (Valanginian and Lower Oxfordian) to 10 m in the massive Tithonian carbonates. The complete sequence
is shown in cross-section in Figure 1d. A regional cleavage pervades the stratigraphy, dipping NNW at a low angle (5-10°)

with an average strike of around 210°. Approaching the Morcles Thrust, the cleavage shallows to horizontal and then steepens
to dip at 20-40° towards the SE (Fig. 1b). Throughout the stratigraphy, the finer grained, more thinly bedded units, typically
with higher organic carbon content, exhibit stronger cleavage.





Figure 1: Geology of the Haut Giffre. (a) Location of the Haut Giffre region (red box) within the Alpine chain (dashed lines), on the French-Swiss border. (b) Simplified geological map of the eastern Haut Giffre, including major thrust faults and regional dips. Significant geographic features are labelled. Cross-section line X-Y indicated by dashed line. (c) Main lithological units outcropping in the Haut Giffre. (d) NNW-SSE cross-section as indicated by line X-Y in (a). (e) Stereonets showing cleavage and poles to cleavage at (i) highest stratigraphic levels (Lower Oxfordian and above, limited to Pic de Tenneverge massif), and (ii) lowest stratigraphic levels (within 1km of the Morcles thrust).





The Morcles nappe, in which the study area lies, is the lowermost of the Helvetic nappes, and consists of a 'normal' limb and a lower overturned limb (Ramsay, 1980; Dietrich and Durney, 1986; Dietrich & Casey, 1989; Kirschner et al., 1999; Austin et al., 2008). The normal limb was subject to around 6 km of burial during the Alpine orogeny (Pfiffner, 1993; Kirschner et al., 1999; Austin et al., 2008). It is at this point that peak metamorphic (and hence maximum temperature) conditions are thought to have occurred; these remain 'sub-greenschist' (Kirschner et al., 1995). The overturned limb of the Morcles nappe outcrops in a 600 m-thick band which dips NW, parallel to the Morcles thrust below (Fig. 1d; note that here the Morcles thrust occupies the geometry of a low-angle normal fault). This overturned limb is mostly sheared Tithonian limestone in the study area, with small wedges of other units included in thrust splays. Beneath the nappe, Triassic sands cap the Aiguilles Rouge massif, and these are together treated as basement here.

Regional scale thrust faults in the Haut Giffre cut through multiple carbonate units. The carbonate units themselves have contrasting mechanical properties. Massive or thickly-bedded limestones (e.g. Tithonian, Urgonian) act as competent beams, folding coherently on 100 m-scale wavelengths in the hanging walls and footwalls of thrusts. Interspersed between these massive limestones are a series of thinly-bedded, relatively carbon-rich shales and marls (e.g. Liassic, Lower Oxfordian, Valanginian) which have undergone internal deformation by means of incoherent folding and the formation of multiple internal detachment surfaces. The non-uniform distribution of strain in the Haut Giffre makes it the suitable subject of an investigation into the effect of strain on Raman spectra.

## 3 Sampling strategy

Samples were taken throughout the 3 km thick thrust-stacked sequence. Significant topographic relief in the form of inaccessible cliff sections necessitated sampling at laterally distributed sites (Fig. 1b); these sites are represented in the cross-section (Fig. 1d) as lateral equivalents. Sample sites that could not be traced laterally to the section line, e.g. due to faulting or folding, are not included in the study.

Samples were categorised for their level of strain and classified on a simple binary scale, as either (1) being distal from thrusts or shear zones, where strain fabrics were present but not intense, indicative of a background level of strain; or (2) where zones of intense strain were present. These two sample site types are termed (1) "background" and (2) "strained" for the purpose of this study. It should be noted that although all samples have been subject to regional deformation, the term "strained" in this context implies that the samples have undergone localised deformation in the form of thrust faults or shear zones, as opposed to "background" strain levels.

At all sample sites, field observations of strain fabrics were made, to inform decisions as to whether a sample would be classified as 'strained' or 'background'. These structural field observations were later compared with microstructural observations using optical microscopy. At strained sites, fine-scale (0.1-10 m) sample transects were made across the thrust faults and shear zones, into the surrounding, less intensely strained material. Background samples were collected to establish the trend in parameters through the complete stratigraphy. At these sites, transects were not made and background sample sites represent one sample or the average of a cluster of 2-3 samples to capture any heterogeneity.











Figure 2 (previous page): 'Strained' localities. (a) Tenneverge thrust, showing (i) Tithonian limestone in the hanging wall thrusted at a low angle over Valanginian shale, and (ii) a close-up of the thrust surface, showing penetration of the fault zone into the footwall shales. (b) Salvadon thrust, showing (i) the thrust surface at outcrop scale, with Tithonian limestone thrust over Valanginian shales, and the direction of displacement obliquely out of the page; and (ii) a close-up view of the thrust surface, showing a clear step change in lithology and small undulations in the surface. (c) Finive shear zone and thrust splay, with (i) the fault exhibiting displacement on a 1 to 10 m scale, occupying the apparent geometry of a low-angle normal fault; (ii) ductile fabrics of the surrounding shear zone. (d) Emaney shear zone, demonstrating (i) the partitioning of strain into the overlying shales, evident in the differing fabric intensities; (ii) a close-up of the boundary between the two units.



### 3.1 Background sample sites

Samples were deemed to be subject to background strain only, if they conformed to two criteria: (1) strain fabric at the sample site was parallel to the regional strain fabric in that area, and (2) the sample strain fabric was visually interpreted to be of similar intensity to the regional fabric in that unit. This interpretation, for the purposes of initial sample selection, was based on field observations; confirmed through inspection of micro-scale structures in thin section. Practically, this meant that the sample was not part of a shear zone or a fault. Background samples were collected at distances of greater than 10 m from such localised high-strain zones. Where a high-strain zone was diffusely bounded, with a gradual return to background levels, the area was avoided entirely for the purpose of background sampling. Since the entire field area is part of a fold-thrust system, avoiding localised zones of high strain significantly limited potential sample sites. In total, 22 background samples from 15 different sites were included in the study, distributed approximately evenly within the intervening stratigraphy, between strained sites.

### 3.2 Strained site samples

Four "strained" sample sites were selected (Fig. 2), three of which are centred around thrusts (Tenneverge, Salvadon, and Finive), and one in the Emaney shear zone. They are described in detail below. Displacements across thrusts and shear zones are estimated from cross-sections based on field mapping.

### 3.2.1 Tenneverge (Fig. 2a)

The Tenneverge thrust forms a discrete fault plane between Valanginian in the footwall and the Tithonian which overlies it (Fig. 2a(i)). Displacement here is estimated at 1 km (Fig. 1d). The Tithonian in the hanging wall is overturned at this locality, and the thinner beds at the base of the Tithonian sequence, which here lie (overturned) directly above the thrust plane, show localised tight chevron folding.

Samples collected from the Valanginian footwall at 12 m, 5 m and 0.5 m from the thrust plane show a gradual reduction in observable primary bedding towards the thrust, replaced by a deformation fabric of increasing intensity (Fig. 2a(ii)). This intensification is manifested in a transition from visible bedding planes, coupled with the regional sub-horizontal deformation fabric, into a dominant fault-parallel foliation which overrides the other fabrics. In the final 0.5 m below the thrust surface, the deformation fabric has been further deformed by rotation and small detachments, suggesting highly localised strain partitioning as part of a complex evolutionary history involving multiple fault movements and fabric overprinting. Intense veining accompanies this deformed layer (Fig. 2a(ii)). In the 10 cm adjacent to the thrust plane the fabric appears more coherent, with the foliation orientated parallel to the fault surface.





### 3.2.2 Salvadon (Fig. 2b)

The Salvadon thrust is a regional thrust fault with a maximum lateral displacement of 2 km towards the NW (Fig. 1d). At the sample site, Tithonian limestone is thrust over a thickened wedge of Valanginian shale. The thrust plane dips approximately 25° SE, with Tithonian bedding sub-parallel to this (strike 122°, dip 25° SW). The contrasting competencies of the lithologies here give rise to a discrete fault surface (Fig. 2b(i)).

  Small (10 cm-scale) undulations in the fault surface (Fig. 2b(ii)), along with small fractures interrupting bedding at the base
of the Tithonian, indicate that some strain was partitioned into the section of the Tithonian most proximal to the thrust (<0.5 m thick). Above this, the Tithonian loses evidence of additional horizontal strain, reverting to bedding-parallel stylolites and orthogonal sub-vertical fracture sets, common to the Tithonian throughout the Haut Giffre. In the footwall, Valanginian shales show evidence of increased strain several metres below the thrust surface. The first metre below the fault plane is dominated by a fault-parallel foliation (050/23° SE), which gradually rotates towards a more bedding-parallel orientation (122/25° SW)
with distance from the fault surface over ~4 m. In a 4-5 m thick zone approaching the fault, en-echelon and orthogonal fracture sets (Fig. 2b(i)) are present in the footwall, indicative of a high degree of strain.. Foliation-parallel veins are also present in the upper 1m of the footwall, increasing in frequency towards the thrust plane (Fig. 2b(i)).

### 3.2.3 Finive (Fig. 2c)

  The Finive sample site is an intraformational thrust splay (Fig. 2c(i)). The splay branches from a regional thrust below, which
separates the sheared overturned lower limb of the Morcles Nappe from the normal limb above, and runs parallel to the Morcles thrust 400 m below (Fig. 1d). Above this regional thrust, Bajocian marls and Liassic shales are tightly folded and thrusted. All samples at the Finive sample site are from the lowermost Bajocian, within which the intraformational thrust splay sits, and it is likely that this portion of the unit is overturned. However, most sedimentary features here have been heavily, if not fully, overprinted during deformation. Compositional layers have been stretched and thinned to 1-5 cm (Fig. 2c(ii)), around 10-30%
of the thickness of such layers outwith this deformation zone. Boudinage and cm-scale folding are both common features in these compositional bands. Straight, foliation-parallel veins 1-5 mm thick are a pervasive feature. The thrust splay, around which sampling was concentrated, is parallel to the deformation fabric which is at a low angle to compositional layering (around 10° separation); as a result, displacement is difficult to estimate. Samples were taken from 10 m above, 0.1 m above, 0.1 to 1 m below, 2 m below, and 10 m below the thrust.

### 3.2.4 Col d'Emaney (Fig. 2d)


  Samples taken from Col d'Emaney are from the base of the sheared lower limb of the Morcles nappe. Here, a wedge of Bajocian material overlain by a shale-rich unit (Fig. 2d(i)) is overthrust by overturned Tithonian. It is unclear as to whether the Bajocian wedge is overturned or not, as the overlying shale-rich unit is highly sheared. Previous geological surveys (e.g. BRGM) have mapped the shale as Oxfordian (suggesting this wedge is the right way up) but could also be overturned Liassic;





distinction based on field observations is inconclusive, owing to the strong strain fabric that overprints sedimentary characteristics in the shale. The precise lithological unit is not of great importance here; what matters is the position within the overall thrust-stacked sequence, and the mechanical properties of the unit.

The contact of the shale unit with the Bajocian gives rise to a 4 m-thick shear zone within the shale, where deformation fabrics are greatly enhanced. S- and C-style fabrics are visible on a cm-scale, along with low-angle fractures which tend to run parallel

to the shear fabrics and are bounded by rotated compositional bands (Fig. 2d(ii)). The combination of these ductile fabrics and brittle, blocky fractures suggests a complicated deformation history. Many of the S-C shears form tight clusters which act to increase the discontinuity between 'blocks' of material. Additionally, deformation-related undulations in the upper surface of the Bajocian (which resemble 0.5 m-scale normal faults, with rotation of the Bajocian cleavage to run parallel to the lithological contact) are accompanied by significant concentrations of ductile strain fabrics and veining in the shale above (Fig. 2d(ii)).

The entire wedge is sheared to an extent, evidenced by strong cleavage and thinning of compositional bands. However, approaching the contact (i.e. within 1 m), the Bajocian cleavage rotates to be almost parallel to the perturbations in the contact surface, resulting in metre-scale 'waves' in the fabric. Within the shale, the highly concentrated deformation zone extends for around 4 m before the fabric consistently returns to the regional orientation.

## 4 Raman spectroscopy and temperature calculations

### 4.1 Introduction to Raman spectroscopy

Raman spectroscopy measures the wavelengths of backscattered radiation produced by different forms of organic carbon. In rocks, organic carbon can take on a range of nanostructures, depending on the peak temperature and strain conditions to which the rock has been exposed. Initially, the carbon will exist in the form of fossilised organic material, whose nanostructure represents that of kerogen (Thrower, 1989; Beyssac et al, 2002a; Rouzaud et al., 2015). As temperatures start to increase, the

carbon nanostructure breaks down into smaller fragments as bonds are broken. With the application of strain (Kwiecinska et al., 2010; Kitamura et al., 2012; Savage et al., 2014; Furuichi et al., 2015; Kitamura et al., 2018) or very high temperatures (Wopenka and Pasteris, 1993; Beyssac et al., 2002b; Schito et al., 2017), these fragments are aligned into parallel sheets, pertaining towards a graphitic nanostructure. Graphitisation has not occurred in any samples in this study.

### 4.2 Sample preparation and spectral acquisition

A total of 62 samples were crushed and powdered before being treated with HCl to remove excess inorganic carbon and therefore improve the signal to noise ratio when obtaining Raman spectra (Pasteris, 1989; Salver-Disma et al., 1999; Beyssac et al., 2002b; Mostefaoui et al., 2008; Muirhead et al., 2012). The residue was then rinsed and dried at room temperature to avoid thermal alteration. Using a Renishaw InVia Raman Spectrometer at the University of Aberdeen, a 514 nm laser was targeted at individual grains in the residual powders, where the laser power was <3 mW at the sample, and spot size was 1-2

μm. Each run comprised three lots of 5 second acquisitions, carried out on 10 individual grains from each sample.




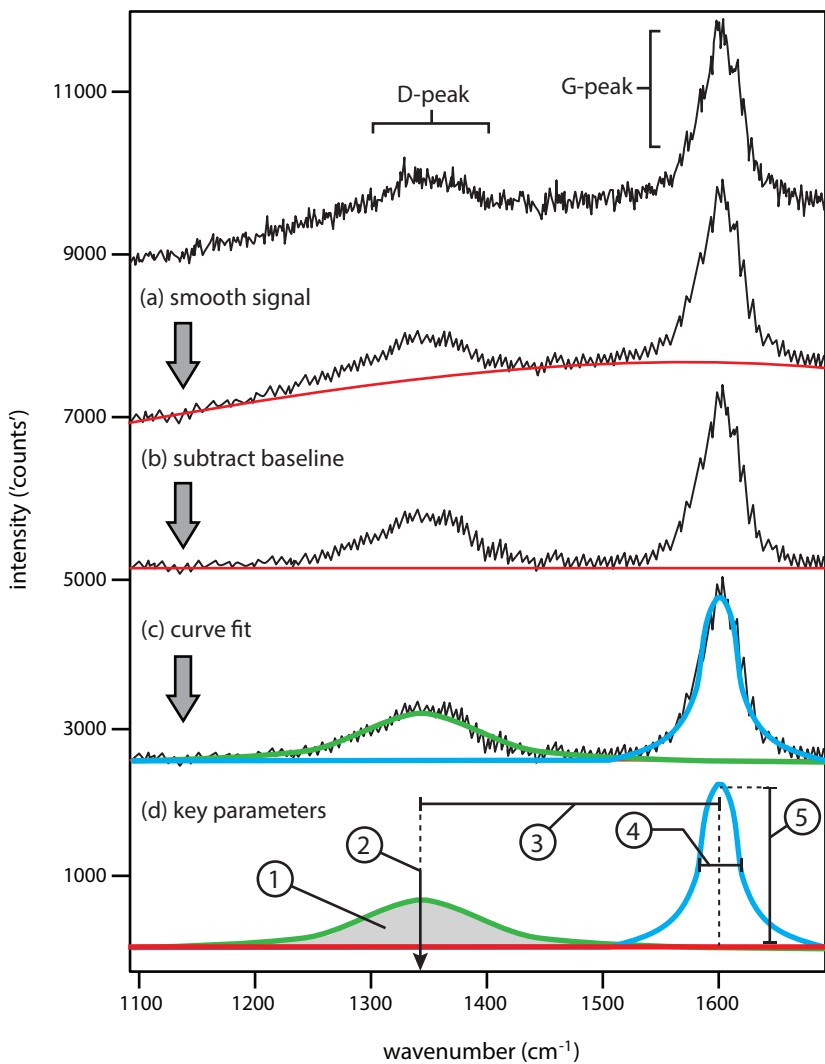

Figure 3: Deconvolution process using WiRE 3.4, showing (a) noise reduction, (b) baseline removal by means of user-guided cubic spline interpolation, (c) Gaussian-Lorentzian hybrid fitting of two curves, and (d) key components used in parameter calculations: (1) = curve area (A), (2) = peak position (W), (3) = Raman Band Separation (RBS), (4) = full width at half maximum (FWHM), (5) = peak intensity (I).





Backscattered radiation was recorded, deconvolved, and analysed using Renishaw WiRE 3.4 software (Fig. 3). Using in-built software functions, noise reduction was first carried out on each spectrum (Fig. 3a), before baseline removal was performed using a cubic spline interpolation which was user-guided (Fig. 3b). Finally, a Gaussian curve fit was applied to the two visible peaks in the spectrum (Fig. 3c), as in Bonal et al. (2006), and the Raman spectra parameters (peak intensity, position, width

and area) were recorded (Fig. 3d; Quirico et al., 2009; Olcott Marshall et al., 2012). This process was carried out 3 times for each acquired spectrum, resulting in 30 analyses per sample. Data points presented in this study therefore represent an average derived from 30 spectra per sample.

## 4.3 Raman spectral parameters

Different Raman spectral parameters are used in combination to determine the carbon nanostructure in a sample. Figure 3d
highlights five key spectral parameters that can be calculated from the two curves fitted to the D- and G-peaks:

(1)   Peak area (A). The height and therefore area of a single peak is affected by signal strength, but comparing the areas beneath the D- and G-peaks negates this issue. The most common ratio comparison is known as 'R2', which is calculated as $A[d]/(A[g]+A[d])$.

(2)   Peak position (W). This is the wavenumber of the peak. In this study we consider a broad D-peak around
1350 cm$^{-1}$, and a sharp G-peak in the range of 1585-1610 cm$^{-1}$.

(3)   Raman Band Separation (RBS). This is the difference between the two peak positions ($W[g]-W[d]$).

(4)   Peak width (FWHM). Calculated as the 'Full Width at Half Maximum', FWHM is measured parallel to the horizontal axis.

(5)   Peak intensity (I). The intensity of a single peak is a direct product of signal strength, so it is more
common to use the ratio between the D- and G-peaks ($I[d]/I[g]$).

The G-peak is in fact a composite of three spectral bands, but these are indiscernible at low metamorphic grades, such as those in this study. For that reason, we collectively refer to these bands as one peak, the G-peak.

Figure 4 shows a schematic summary of the changes in Raman spectral parameters with increasing temperature and strain. At low maturities, as kerogen-like carbon degrades with increasing temperature, the D-peak increases in intensity (Tuinstra and
Koenig, 1970; Beyssac et al., 2002). This increases the $I[d]/I[g]$ ratio (Fig. 4a(i)). Subsequently, as higher maturities are reached, carbonaceous fragments align into sheets and the G-peak becomes more intense, decreasing $I[d]/I[g]$ (Muirhead et al., 2012, 2017; Buseck and Beyssac, 2014). Generally, a decrease in $I[d]/I[g]$ is observed when strain is applied to relatively low-maturity organic carbon (Fig. 4a(ii)); Kwiecinska et al., 2010; Kitamura et al., 2012; Furuichi et al., 2015), but brittle fragmentation has been reported when mature, near-graphitic carbon is subject to low temperature strain, resulting in an
$I[d]/I[g]$ increase (Nakamura et al., 2015; Kirilova et al., 2018).

In addition to changing intensity, both the D- and G-peaks shift towards lower wavenumbers as the material approaches pure graphite, but the D-peak shifts more significantly (Wopenka and Pasteris, 1993; Beyssac et al., 2002; Quirico et al., 2009).



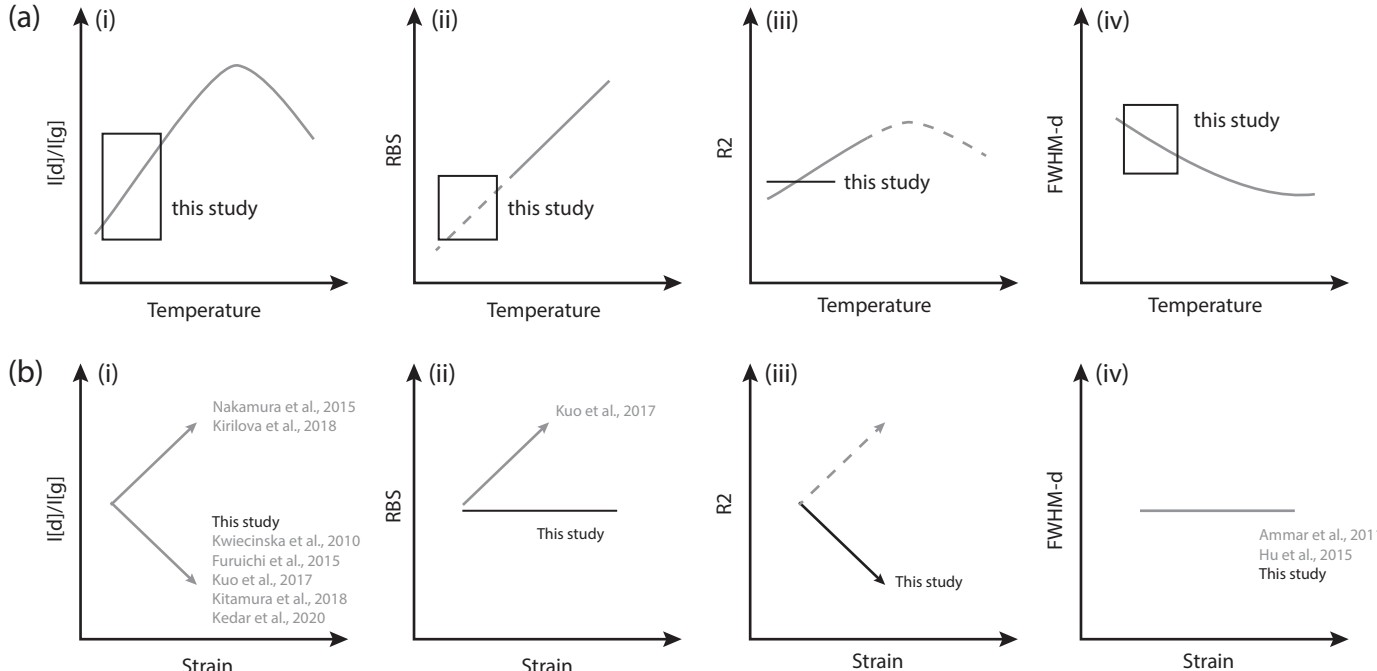

Figure 4: Typical changes in Raman spectroscopic parameters with temperature and strain, represented schematically. Exact temperature, strain, or parameter values are not stipulated, since these will vary depending on starting material. Where data from this study matches the trend, the applicable zone is marked with a black box; where it differs from the usual trend, data from this study is marked with a black line representing the observed trend. (a) Parametric changes with temperature: (i) Intensity ratio (I[d]/I[g]); (ii) Raman Band Separation (RBS); (iii) area ratio (R2); (iv) D-peak width (FWHM-d). (b) Direction of parametric shifts with the application of strain: (i) intensity ratio; (ii) Raman Band Separation; (iii) area ratio; (iv) D-peak width.




This causes the RBS to change (e.g. Zhou et al., 2014; Schmidt et al., 2017; Schito and Corrado, 2018; Henry et al., 2019). Figure 4b(i) shows an increase in RBS with increasing temperature at higher maturities, but little is known about how the
parameter changes at low maturities such as in this study. The effect of strain has not been studied extensively, although Kuo et al. (2017) reported an increase in RBS with the application of strain (Fig. 4b(ii)).

Figure 4c(i) illustrates how the R2 (curve area) ratio has been shown to correlate with temperature (Beyssac et al., 2002; Aoya et al., 2010; Nakamura et al., 2015; Chen et al., 2017; Kirilova et al., 2018; Henry et al., 2019). It follows a similar pattern to that of I[d]/I[g] with temperature, but little work has been done to establish whether this similarity extends to strain (Fig.
4c(ii)).

It is generally accepted that the width (measured as full width at half-maximum, FWHM) of the D- and G-peaks change with temperature (Zeng and Wu, 2007; Aoya et al., 2010; Kouketsu et al., 2014; Zhou et al., 2014; Hu et al., 2015; Bonoldi et al., 2016; Chen et al., 2017). However, the nature of the change varies depending on the thermal and barometric conditions, along with the nature of the organic starting material. At relatively low temperatures (<300°C), these studies report a decrease in
FWHM-D with increasing temperature (Fig. 4d(i)), which also correlates with an increase in I[d]/I[g]. FWHM-D is thought to undergo very little change, if any, when exposed to differential strain (Ammar et al., 2011; Hu et al., 2015; Fig. 4d(ii)).

Therefore, comparison of the relative intensities, positions, widths, and areas of the D- and G-peaks is a common method of assessing the extent to which a rock has been heated, which can be correlated to geological processes such as burial depth, contact metamorphism, and exposure to hot fluids. By comparison, the investigation of the effect of strain on these parameters
is in its infancy (Kwiecinska et al., 2010; Kitamura et al., 2012; Furuichi et al., 2015; Kuo et al., 2017; Kedar et al., 2020).

**4.4 Temperature calculations**

Using the parameters described above, three different methods of calculating temperature from Raman data were employed. Choice of method was based on the assumption of regional peak temperatures of less than ~300°C in the field area, due to evidence of low-temperature metamorphism not exceeding sub-greenschist facies (Kirschner et al., 1995). This excludes
certain Raman-based geothermometric equations which are only applicable above 350°C (e.g. Wopenka and Pasteris, 1993; Beyssac et al., 2002; Aoya et al., 2010).

The first equation used was that of Schito and Corrado (2018), which uses (in order of decreasing significance) the following Raman spectral parameters: I[d]/I[g] (intensity ratio), RBS (Raman Band Separation), FWHM[d] (D-peak width), FWHM[g] (G-peak width), A[d] (D-peak area) and A[g] (G-peak area) to give a %Ro equivalent between 0.3 and 1.0. This is then
converted to an approximate temperature value using the equation proposed by Barker and Pawlewiscz (1994):

$$T_1 = \frac{\ln(\%Ro_{eq}) + 1.68}{0.0124}$$

The second temperature calculation used is that of Kouketsu et al. (2014). This equation is based purely on FWHM[d] (D-peak width), and is reported as being effective in the range 150°C < T < 400°C:

$$T_2 = -2.15(FWHM_D) + 478$$





The final temperature calculation used was that proposed by Lahfid et al. (2010), which converts R2 (area ratio) into temperature using the following equation:

$$T_3 = \frac{R2 - 0.3758}{0.0008}$$

where

$$R2 = \frac{A_D}{A_G + A_D}.$$

Lahfid et al. (2010) report that their equation is most accurate within the range 200°C < T < 320°C.

## 5 Raman spectral parameters across the fold-thrust belt

Figure 5 shows four individual Raman spectral parameters (I[d]/I[g], R2, FWHM[d] and RBS) plotted on separate cross-sections, with each point corresponding to a sample site. Background samples taken distal from thrusts and shear zones are marked as single points. We first consider the relationship between the thrust plane and shear zone samples and their proximal

neighbours, and how these relate to the regional trends observed across the area and visualised on the cross-section. Later we examine in more detailed the 10 cm to 1 m-scale sampling in the interceding 10 m above and below the thrust planes and across the shear zones. Results are grouped by parameter, in each case with the "background" samples described first, followed by the "strained" samples.

### 5.1 I[d]/I[g]

I[d]/I[g] shows a general trend from lower values (0.3-0.4) in the upper stratigraphy to higher values (0.7-0.8) in the lowest stratigraphy (Fig. 5a) in background samples. This trend correlates with the depth through the thrust stack, and the graph of I[d]/I[g] with depth (Fig. 5a(ii)) highlights the gradual increase towards higher values approaching the basal thrust. The Morcles thrust flattens towards the NW end of the cross-section, coinciding with a more vertical trend in I[d]/I[g] values. This trend appears to be disrupted across thrust planes and shear zones in the strained samples. In the case of the Salvadon thrust,

the samples taken 10 m above and below the thrust plane are consistent with the regional trend, with values of 0.377 and 0.417 respectively. However, I[d]/I[g] values on the thrust plane are 25-30% lower, with an average value of 0.249. There is a similar though slightly less significant drop on the Tenneverge thrust plane, from 0.552 and 0.541 to 0.469, a decrease of around 18%. On the Finive thrust plane there is a drop to 0.632 from surrounding values of 0.739 and 0.772. The Emaney shear zone, which is not immediately associated with a major fault, exhibits values of 0.624 and 0.627 within the shear zone itself, with higher

values (0.772 and 0.888) outwith the shear zone, similar to the thrust localities.

### 5.2 RBS

Raman band spacing (RBS) appears to show no consistent pattern through the stratigraphy (Fig. 5b), apart from a prevalence of values >265 in the upper stratigraphy and <265 in the lower sequence, but the average error associated with each sample is









Key for Figure 5:

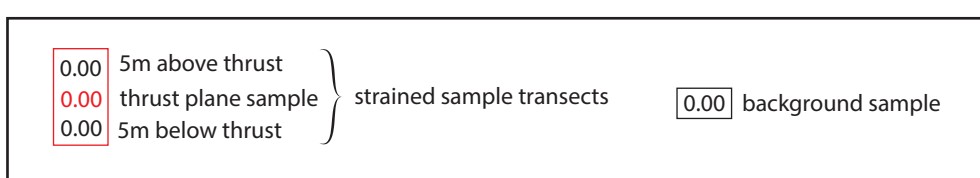

Figure 5 (previous page): Cross sections and depth transects showing values of (a) I[d]/I[g], (b) Raman band separation, (c) area ratio R2, and (d) D-peak width. Labelled values on cross-sections indicate approximate position of samples, with numbers in black boxes representing 'background' samples and those in red boxes indicating samples through 'strained' localities. Within strained localities, red coloured numbers represent fault plane or shear zone samples. The depth transects on the right of each panel are accompanied by a simplified stratigraphic column for reference. Strained localities are highlighted by red boxes labelled with a corresponding letter (T = Tenneverge, S = Salvadon, F = Finive, E = Emaney), with the thrust plane or shear zone sample indicated by a red dot.





+/- 4, suggesting that the change through the stratigraphic sequence is not significant. Unlike I[d]/I[g], there is little change on
the thrust planes; however, there is a small shift to slightly higher values in the Emaney shear zone (263/261 from 255/259
below the shear zone), but the high error causes these ranges overlap.

### 5.3 R2

R2 (the area ratio; Fig. 5c) shows little overall change with depth towards the Morcles thrust, although there is a weak trend
towards higher values with increasing depth within individual thrusted packages. There is a marked drop on each thrust plane
and within the shear zone, similar to the behaviour seen in I[d]/I[g]. On the Salvadon thrust, there is a decrease from 0.572 (10
m below the thrust plane) and 0.568 (10 m above the thrust) to 0.460 on the thrust plane. On the Tenneverge thrust plane the
R2 value is 0.486, compared to 0.502 above and 0.518 below. In the case of the Finive thrust, R2 is 0.512 on the thrust plane,
whilst 10 m above and below it, R2 sits at 0.580 and 0.555 respectively. Finally, there is also a decrease in R2 moving from
10 m below the Emaney shear zone (0.560 and 0.564) to within it (0.549 and 0.529). The average error for each sample was
+/- 0.078, so only the Salvadon thrust samples exhibit a change with a greater magnitude than this, but importantly the direction
of change is consistent.

### 5.4 FWHM[d]

Only FWHM[d] is shown in Fig. 5d, as FWHM[g] varies significantly and shows no discernible trend in our data (the reader
is referred to Supplementary Material). There appears to only be a small shift, if any, on thrust planes and in the shear zone.
There is, however, a general decrease in FWHM[d] with depth towards the Morcles thrust (Fig. 5c) and therefore a reverse
correlation with I[d]/I[g], as illustrated in Figure 6.

### 6 Temperature calculations across the fold-thrust belt

Using the above parameters, the results of Equations 1, 2 and 3 (from Schito and Corrado (2018), Kouketsu et al. (2014), and
Lahfid et al. (2010), respectively) were plotted on the cross-section for each sample site.

### 6.1 Temperatures based on Schito and Corrado (2018)

Applying the Schito and Corrado (2018) equation – based on I[d]/I[g], RBS, FWHM[d], FWHM[g], A[d] and A[g] – to our
data gives a calculated temperature range across all the samples of 79°C to 104°C (Fig. 7a). In background samples within
intra-fault stratigraphic packages, there is a slight trend towards higher temperatures with increased depth towards the Morcles
thrust. The calculated temperatures appear to show a 10-20°C increase as thrusts are approached. There is little change on
thrust planes compared to surrounding values, with the only significant change being that of the Salvadon thrust, where the
thrust plane 'temperature' is calculated at 93°C compared to 100°C and 103°C above and below it. In contrast, there is a slight
increase in calculated temperature in the Emaney shear zone (85 and 88°C) compared to below it (79 and 81°C).





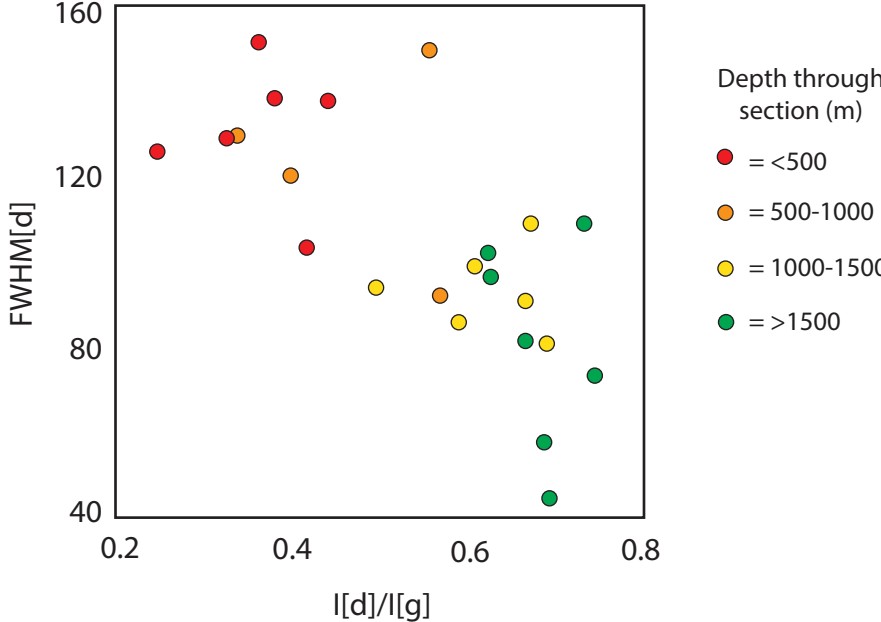

Figure 6: D-peak width (FWHM-d) plotted against intensity ratio (I[d]/I[g]), with points coloured by depth through the overall transect.









Figure 7 (previous page): Cross sections and depth transects showing results of temperature calculations using the equations proposed by (a) Schito & Corrado 2018, (b) Kouketsu et al., 2014, and (c) Lahfid et al., 2010. Labelled values on cross-sections indicate approximate position of samples, with numbers in black boxes representing 'background' samples and those in red boxes indicating samples through 'strained' localities. Within strained localities, red coloured numbers represent fault plane or shear zone samples. The depth transects on the right of each panel are accompanied by a simplified stratigraphic column for reference. Strained localities are highlighted by red boxes labelled with a corresponding letter (T = Tenneverge, S = Salvadon, F = Finive, E = Emaney), with the thrust plane or shear zone sample indicated by a red dot.





### 6.2 Temperatures based on Kouketsu et al. (2014)

Since the Kouketsu et al. (2014) equation relies solely on FWHM[d], the calculated temperatures show a similar (although reversed) pattern to that produced by the FWHM[d] parameter itself. There is a slight overall trend towards higher temperatures towards the basal thrust (290-320°C compared to 250-280°C in the upper section), but the pattern is irregular and a clear trend is difficult to identify. There is a drop of around 30°C on the Tenneverge thrust plane, and 5-10°C in the Emaney shear zone (Fig. 7b). However, the Salvadon and Finive thrust planes do not show significant change, and fall within the variation of temperatures calculated for the samples above and below those thrusts.

### 6.3 Temperatures based on Lahfid et al. (2010)

The Lahfid et al. (2010) equation calculates temperatures as a linear derivative of R2, and so the trend is similar to that of R2 but easier to discern due to the larger range of values (138-312°C). There is a clear increase in temperature both with depth and with proximity to the basal Morcles thrust (Fig. 7c). There is disruption in the vicinity of regional-scale folds and thrusts; notably, temperatures above and below the Salvadon thrust are significantly higher (240°C) than those at similar stratigraphic levels (150-190°C). On all three thrust planes and within the shear zone, there is a marked decrease in calculated temperature, ranging from a 20°C difference (Tenneverge) to 140°C (Salvadon) below the temperatures calculated for the surrounding samples. As with the Schito-Corrado temperature calculation, there is a slight increase (10-30°C) in apparent temperature approaching the thrusts compared with 'background' values.

## 7 Discussion

### 7.1 I[d]/I[g]

At low thermal maturities, increasing temperatures cause a rise in I[d]/I[g] (Fig. 4a; Dietrich and Casey, 1989; Muirhead et al., 2012; Sauerer et al., 2017). In the Haut Giffre, we estimate peak burial temperatures to be 150-250°C, based on a standard geothermal gradient of 25°C/km and an estimated burial depth of 6km at the top of the exposed stratigraphic pile (Pfiffner, 1993; Kirschner et al., 1999; Austin et al., 2008). Therefore, it is reasonable to suggest that the increase in the I[d]/I[g] ratio towards the basal Morcles thrust is associated with increasing peak temperature, and hence maximum burial. This fits with the observations, for example, of Schito et al, (2017), where a similar trend of increasing I[d]/I[g] with depth is seen through some 4 km of core containing siliciclastics from the Lower Congo Basin, Angola, at temperatures up to 170°C.

There is a significant drop in I[d]/I[g] on thrust planes and in shear zones, with as much as a 40% decrease in the ratio values on the Salvadon thrust plane compared to the surrounding stratigraphy (0.417 to 0.279, a difference of 0.168) and 30% difference between samples in the Emaney shear zone to those adjacent (0.888 to 0.624, a difference of 0.264). There are several possible causes of this drop:





(1) A lower peak temperature on the fault plane than the surrounding rock would reduce the I[d]/I[g] value on the fault plane. However, there is no plausible mechanism to explain how this would occur.

(2) A very large temperature increase on the fault plane (>500°C) could cause such a spectral change that I[d]/I[g] values begin to drop again (Fig. 4a; Bustin et al., 1995; Furuichi et al., 2015; Kaneki et al., 2016). A possible mechanism for such a local temperature elevation could be flash heating due to friction on the fault plane. Frictional heating is known to occur on fault planes (Goldsby and Tullis, 2007; Smith et al., 2015), particularly in episodes of rapid seismic slip (Rabinowitz et al., 2020). However, the magnitude and duration of elevated temperatures from friction depend on a range of factors such as permeability, slip duration, and fault thickness (Bustin, 1983; Mase and Smith, 1987; Fulton and Harris, 2012; Kitamura et al., 2012), and there is therefore uncertainty as to whether this would always be sufficient to alter the Raman spectra. Mase and Smith (1987) modelled frictional heating on fault planes and found that in porous rocks, the slip duration would have to be much greater than 100 seconds for thermal pressurisation to occur. Our results show that I[d]/I[g] decreases similarly in thrust faults and in broader shear zones. Transient frictional heating cannot explain the decrease in I[d]/I[g] values in the Emaney shear zone, which has undergone mostly ductile deformation over a more widely distributed area.

(3) Strain-related spectral changes can also reduce I[d]/I[g] (Kwiecinska et al., 2010; Kitamura et al., 2012; Furuichi et al., 2015; Kedar et al., 2020), and this would be applicable to both fault planes and distributed shear zones. Kedar et al. (2020) reported a drop of 0.1 to 0.15 in I[d]/I[g] in the sheared, overturned limb of a recumbent isoclinal fold, corresponding to an increase in strained microfabrics in those samples.

## 7.2 RBS

Raman band separation (RBS) is reported to increase with increasing temperature (Fig. 4b; Zhou et al., 2014; Bonoldi et al., 2016; Sauerer et al., 2017), and so should increase with depth towards the basal thrust in our study. However, there is no visible trend in our RBS data. It is possible that the samples in this study are not of high enough maturity for a trend in RBS to be seen, as some studies only report trends only in samples over 400°C (Sauerer et al., 2017; Henry et al., 2019). If frictional heating on fault planes is the primary control on Raman spectral changes, it would be expected that the maturity would be high enough to enter the zone in which RBS is confirmed to change with temperature (i.e. >400°C). This would correspond to an approximate instantaneous slip magnitude of ~1 m (Savage et al., 2014), which is not unreasonable on the fault planes in question. However, we do not see significant changes in RBS on fault planes or in distributed shear zones compared to the surrounding rocks. This suggests that frictional heating does not play a significant role in changing the Raman spectral parameters on thrust planes,.

## 7.3 FWHM[d]

D-peak width (FWHM[d]) exhibits a reverse trend to that of I[d]/I[g], decreasing slightly with depth towards the basal thrust. This supports experiments by Zeng and Wu (2007), who observed a decrease in FWHM[d] with increasing temperature,





although their experiments were on samples at 300°C and above. Zhou et al. (2014) also observed a decrease in FWHM[d]
with increasing I[d]/I[g] in solid bitumen, similar to the trend seen in this study. Studies on coal approaching a magmatic
contact by Chen et al. (2017) indicate that both FWHM[d] and FWHM[g] decrease with increasing temperature, but in this
study we only observe a decrease in FWHM[d], whilst FWHM[g] changes very little. However, the starting material in this
study is kerogen-like carbon rather than coal, which has a different crystalline structure. This may account for differences
between our study and that of Chen et al. (2017).

Unlike I[d]/I[g], there is little change in FWHM[d] across fault planes or shear zones, which suggests that the two parameters
are not directly related. For samples affected by polishing during sample preparation, it has been noted that Raman spectral
peak widths are less influenced than peak intensities (Ammar et al., 2011; Hu et al., 2015), suggesting that FWHM[d] does not
change significantly due to shearing. In light of our results, it may be possible to extend this suggestion to shearing on fault
planes.

**7.4 R2**

R2 (the area ratio) shows a similar trend to that of I[d]/I[g], only not as pronounced. Note that the range in R2 values is lower
than that of I[d]/I[g] due to the normalised denominator used to calculate R2 (Equation 1), so a weaker trend than I[d]/I[g] is
expected. There is a drop in R2 on thrust planes and within the Emaney shear zone, this drop is 6-20% (a difference of 0.035;
see Fig. 5(c)), compared to 15-40% for I[d]/I[g]. However, the percent change in R2 on fault planes is comparable to the total
change in R2 between the upper and lower sections of the stratigraphy (though this is subject to a high degree of variation).
Since peak area is a product of peak intensity and peak width, it follows that R2 is dependent on I[d]/I[g] and FWHM[d].
These two parameters have opposing trends with depth through the sequence, resulting in a general dampening of any R2
trend. However, in strained samples, I[d]/I[g] tends to drop, whilst FWHM[d] remains unchanged. This means that R2 also
drops, and makes this parameter more sensitive to strain-related spectral changes than to burial trends.

**7.5 Schito and Corrado (2018) calculated temperatures**

The Schito and Corrado (2018) equation uses I[d]/I[g], RBS, FWHM[d] and [g], and individual peak areas to calculate %Ro,
which can be subsequently used to estimate temperature. In this study, temperatures estimated using the Schito and Corrado
(2018) equation sit between 70 and 110°C. This range is lower than expected for rocks that have been buried to 6-9km, as
suggested by previous studies (Pfiffner, 1993; Kirschner et al., 1999; Austin et al., 2008). This could be due to a weak
geothermal gradient, or that the equation is not applicable in this instance due to the Raman parameters used.
The temperature trend calculated using this equation indicates a low geothermal gradient of ~15°C/km. If this thermal gradient
were to be extrapolated upwards through the previous overlying stratigraphy, the range of 70-110°C at 6km depth would
almost be appropriate. It is possible that through thrust-stacking, the regional-scale geothermal gradient could be flattened; if
the majority of thrust emplacement occurred during exhumation, then peak temperatures could have remained relatively low.



The second term in the equation is RBS, which does not exhibit a trend in our data. As a result, its presence in the equation may subdue the range of estimated temperatures. Therefore, at lower maturities, the RBS term in the Schito and Corrado (2018) equation may become irrelevant and make it less effective for temperature determination.

Our data shows a small shift (of the order of 10°C) in temperature on fault planes and in shear zones, but the direction and exact magnitude is inconsistent (as noted previously by Muirhead et al., *in review*). Since the most significant term in the

equation is I[d]/I[g], and our data shows that I[d]/I[g] is strongly affected by strain-related spectral changes, it follows that the equation should be sensitive to strain. Although a strain-induced error in apparent temperature of +/-10°C will not significantly impact the performance of the Raman geothermometer, it highlights the importance of context when estimating temperatures using this method.

A transient temperature rise during frictional heating on a fault plane may be too short-lived to promote spectral changes

(Bustin, 1983; Fulton and Harris, 2012; Kitamura et al., 2012; Furuichi et al., 2015). This may be one explanation for the lack of a consistent temperature increase on the thrust planes in this study. Other explanations include differing thicknesses of active slip (Raboniwitz et al., 2020), or a low slip magnitude in a single event (Polissar et al., 2011; Savage et al., 2014; Savage et al., 2018; Raboniwitz et al., 2020). However, these do not explain the distinct drop in apparent temperature on the Salvadon thrust plane, or the elevated temperature values within the Emaney shear zone, where frictional heating should not play a role.

**7.6 Kouketsu et al. (2014) calculated temperatures**

The Kouketsu et al. (2014) equation for calculating temperature uses just the FWHM[d] parameter, and gives temperatures of 200-380°C. The lower end of this temperature range overlaps with that expected for a burial depth of 6-9 km, but we would not expect to see temperatures above 250°C. Our data suggests that strain should have a limited effect on temperatures derived from this equation, since FWHM[d] is reportedly insensitive to strain (Ammar et al., 2011; Hu et al., 2015).

In the burial trend, there is significant variation on a sub-km scale. In their paper, Kouketsu et al. (2014) highlight an error of +/- 30°C associated with the equation. This magnitude of error, necessitates temperature trends to be identified over km-scale distances or greater for a normal geothermal gradient.

**7.7 Lahfid et al (2010) calculated temperatures**

Using the Lahfid et al. (2010) equation, our data yields apparent temperatures (138-312°C) that are closer to the expected range

for a burial depth of 6-9km (150-250°C). However, the trend shown by these temperatures indicates a high geothermal gradient (70-80°C/km) that is not easily explained through thrust tectonics.

Since the Lahfid et al. (2010) equation is entirely dependent on R2, it follows that strain will significantly affect the results. This fits with a consistent drop of 40-50°C observed on fault planes and in shear zones within the study area.



## 7.8 Summary

From our observations we suggest that the Schito and Corrado (2018) equation is less effected by strained environments than the Lahfid et al. (2010) equation. The Kouketsu et al. (2014) equation is also more suited to strained environments. However, the Schito and Corrado (2018) equation produces temperature estimates and a geothermal gradient lower than expected for the region and the Kouketsu et al. (2014) equation, in our case study, shows variation in temperature predictions on a sub-km scale, making it less suitable for general use. Unlike the Schito and Corrado (2018) equation, the Lahfid et al. (2010) equation

demonstrates a more consistent error in the most strained rocks, and the predicted temperatures are more in line with those predicted for the area. The consistency of the shift on thrust planes and in shear zones with the Lahfid et al. (2010) temperature calculation suggests that it might be possible to correct for this with contextual sample knowledge, or by comparison with other equations.

## 8 Conclusions

Analysis of samples from an Alpine carbonate fold-thrust system has revealed trends and anomalies in Raman spectral data. We chose four key parameters which are frequently used to assess thermal maturity of organic carbon in rock samples, and plotted the values at the corresponding sample sites on a cross-section. By separating samples that had been affected by locally high strain (such as on fault planes or in shear zones) from those that had only been subjected to the background regional strain, we were able to apply context to the data and hence discern regional thermal trends from localised strain-related

anomalies.

Parameters showed varying sensitivities to strain and temperature. In background samples, I[d]/I[g] increased with depth towards the basal thrust, suggesting an expected 'burial trend'. FWHM[d] decreased with depth, whilst R2 – a product of I[d]/I[g] and, to some extent, FWHM[d] – increased slightly. RBS showed no discernible trend. In strained samples, I[d]/I[g] dropped by 0.1 to 0.15 (up to 40% depending on location in the stratigraphy), and R2 showed a small decrease. There was

little change, if any, in FWHM[d] or RBS in strained samples.

## 8.1 Calculated temperature trends

Spectral data from the samples in this study were applied to three different temperature equations developed by Schito and Corrado (2018), Kouketsu et al. (2014) and Lahfid et al. (2010) respectively.

Between thrusts, the Schito and Corrado (2018) equation produced a broad trend of weakly increasing temperatures with depth.

Overall, these temperatures were lower than expected. The Lahfid et al. (2010) equation produced temperatures which were within the expected range for 6 km of burial, but also indicated a very high geothermal gradient. The Kouketsu et al. (2014) equation gave temperatures which were higher than expected for the proposed burial depth, with only a slight trend towards higher values at depth, and a high degree of variation. This variation suggests that the Kouketsu et al. (2014) equation is unsuitable for establishing temperature gradients over distances of less than a few km. In fold-thrust systems such as the one





investigated here, in which calculated temperatures may be influenced by strain, an equation is required that can resolve
temperature changes over hundreds of metres at the least. We conclude that although various geothermometric equations carry
applicable temperature ranges, choosing the most appropriate equation is complex and dependent on multiple factors.

## 8.2 Impact of strain on calculated temperatures

Choice of equation is particularly important in the context of fold-thrust systems, where strain intensity can vary on multiple
scales. The use of multiple parameters in the Schito and Corrado (2018) equation suggests that the equation should be relatively
insensitive to strain. However, on thrust planes, the Schito and Corrado (2018) calculated temperatures dropped by 0-10°C,
while increasing by the same magnitude in a distributed shear zone. This suggests that the equation is indeed sensitive to strain-
related spectral changes, likely due to the fact that I[d]/I[g] is the dominant term in the equation. The use of multiple terms in
the equation may help to produce more reliable results (as the influence of different parameters counteract), it is important to
consider which parameters may have the most influence in different areas and indeed different samples.

The results of the Kouketsu et al. (2014) equation showed considerable variations in temperature, with no consistency when
moving from background to locally strained samples. Therefore, despite being based on a parameter which is, in theory,
relatively unaffected by strain, temperatures produced by this equation appear unreliable in this case. The Lahfid et al. (2010)
equation recorded a consistent drop of 40-50°C in strained samples, suggesting that this temperature equation is significantly
affected by strain-related spectral changes. The consistency of this calculated temperature drop suggests that it might be
feasible to compare these results with those of another equation, and/or the geothermal gradient to distinguish the effect of
strain-related spectral changes from those induced by temperature. This finding is important not only for improving the
robustness of Raman spectroscopy as a geothermometer in fold-thrust systems, but also for the potential to develop a Raman-
based strain tracker.

**CRediT Author Statement**

**Kedar**: fieldwork, Raman spectroscopy analysis and interpretation, original draft preparation, figure preparation; **Bond**:
original conceptualisation, input into rewriting and framing original draft, fieldwork (support); **Muirhead**: Raman
spectroscopy interpretation, input into writing and re-drafting of original draft.
All authors have contributed to the writing and framing of the manuscript and discussion of all concepts.

**Declaration**

The authors declare that they have no conflict of interest.



**Acknowledgements**

This study was carried out as part of a University of Aberdeen PhD, supported by the UKRI Centre for Doctoral Training in Oil & Gas [grant number NE/R01051X/1].

**Data availability**

Full dataset is available online: Kedar, L., Bond, C. E., & Muirhead, D. (2021). Raman spectral data from carbonates in the Haut Giffre, French Alps [Data set]. Zenodo. http://doi.org/10.5281/zenodo.4771951

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
