# Peer review of "Raman spectroscopy in thrust-stacked carbonates: an investigation of spectral parameters in strained samples"

_Solid Earth, 2021_

## Author Comment (AC1)

Authors' response:

Dear Dr. Jubb,

Thank you for taking the time to study our manuscript and for writing such a helpful and comprehensive review. We have taken your comments very seriously and have, as a result, made significant changes to the manuscript. Our specific responses are detailed below each major and minor comment individually (blue text), but we thought it would be useful to summarise our changes as follows:

- We have focussed the paper on the changes to individual Raman spectral parameters rather than geothermometric equations, only including the geothermometry as a minor discussion point later on in the text. We believe that this makes the manuscript far clearer and more focussed, not to mention easier for the reader to surmise the key points of the study.
- Transects across faults and shear zones are presented in more detail.
- Error ranges have been added to both text and figures.
- Methodology and definitions have been significantly tightened.

Once again, we would like to thank you for your time and for your help in improving our manuscript. We believe that with your comments we have been able to make substantial improvements in terms clarity, and we are particularly grateful for your comment summarising the key questions answered by our paper as this had the effect of highlighting those points that we had made clear and those we had not.

Yours sincerely,

L. Kedar, C. E. Bond, and D. Muirhead.

**Major Revisions**

1. Geothermometers were not applied correctly. Specific problems with each detailed below.

Thank you for highlighting your concerns with our use of geothermometers in this manuscript. Upon consideration, we determined that the original manuscript lacked clarity and tried to 'do too much' – we believe that it is more useful to the reader and the scientific community to focus the revised manuscript on the response of individual Raman parameters only, rather than including various geothermometers too. The geothermometry side of the original manuscript could be addressed in a future follow-on paper, which would of course address each of your concerns below.

For each comment below, we have described the changes made, and where appropriate, have explained our reasoning. However, for the most part, the points relating to geothermometric techniques no longer apply to the revised manuscript as we have removed the geothermometry section in order to focus the paper on the individual spectral parameters and their response to strain. This means that the Lahfid and the Kouketsu thermometers are no longer included in this paper; however, the Schito-

Corrado equation is discussed as an implication of our findings relating to individual spectral parameters.

a. Lahfid thermometer: $T = (RA1-0.3758)/0.0008$, here $RA1 = (D1+D4)/(D1+D2+D3+D4+G)$ using peak areas from Lorentzian peaks fit to Raman spectra collected with 514 nm laser. To apply this thermometer correctly you need to:

(i) Collect the Raman data with a 514 nm laser because dispersion of the D peak will change the calculated area. See the following references for discussions on dispersion effects on Raman spectra of geologic organic matter:

Lünsdorf, 2016, Raman spectroscopy of dispersed vitrinite – Methodical aspects and correlation with reflectance, Inter. J. Coal Geol., 153, 75-86.

Sauerer et al., 2017, Fast and accurate shale maturity determination by Raman spectroscopy measurement with minimal sample preparation, Inter. J. Coal Geol., 173, 150-157

Jubb et al., 2018, High microscale variability in Raman thermal maturity estimates from shale organic matter, Inter. J. Coal Geol., 199, 1-9

(ii) Fit the collected Raman data using a sum of Lorentzian profiles as Lahfid did, not the pseudo-Voigt profile that was used here.

Thank you for pointing out this oversight. In future we will bear this in mind; however, we now no longer use the Lahfid equation.

(iii) Fit the collected Raman spectra with five peaks, not two.

Thank you for highlighting these points. A 514nm was used; this has now been specified (Line 240). The original reason for using 2 peaks as opposed to 5 was that some work (e.g. Schito et al., 2017; Schito and Corrado, 2018; Henry et al., 2019) suggests that the two methods produce comparable trends, and so although not absolutely accurate, we reasoned that it would be possible that some authors may try to equate the two methods, and therefore it was reasonable to investigate the results that might be produced using the two-peak approach. We appreciate that this was not made clear in the original manuscript, and any future work will address this problem. The revised manuscript does not include the Lahfid geothermometer any longer as we focus on individual parametric responses.

b. Kouketsu thermometer: $T = -2.15(D1\text{-FWHM}) + 478$, collected with 532nm laser, fit using pseudo-Voigt profiles.

The number of peaks fit to the Raman spectra for the Kouketsu thermometer was dependent on several qualitative and quantitative parameters. See flowchart (Figure 3) from Kouketsu 2014 here:

For the sample temperatures in this study (left end of the flow chart), either 4 or 5 peaks needed to be fit to the Raman data in order to correctly apply this geothermometer.

Once again, thank you for these comments. As with the Lahfid thermometer, the Kouketsu thermometer is not included in the revised manuscript owing to our general

refocussing of the paper. Again, the idea was that general use of such geothermometers might result in cross-overs of methods such as this. However, future work will ensure that correct methodology is followed and that any deviations from this are accompanied by valid reasoning.

c. Schito and Corrado estimation of VRo% for use in the Barker and Pawlewiscz geothermometer: T = (ln(VRo%) + 1.68) / 0.0124.

Here the authors have correctly used relationships from Schito and Corrado to determine an equivalent vitrinite reflectance (VRo%) from the Raman spectra. This parameter is then used to determine a temperature using the 1986 thermometer proposed by Barker and Pawelewicz. However, the data that this thermometer is based on are highly uncertain (see Figure 1 from Barker and Pawelewicz 1986 below), making it semi-quantitative at best. Certainly, correlating vitrinite reflectance to an absolute temperature is a large challenge for the field and one that warrants careful study in its own right. Regardless, for the work reported here, an estimation of uncertainty in the determined temperatures using the Barker and Pawelewicz equation **must** be included if this geothermometer is to be used.

The revised manuscript addresses these points by highlighting the uncertainties described above (Line 488). The Schito and Corrado equation is now primarily used as a discussion point rather than a focus to the paper.

2. Analytical uncertainties need to be added to Figures 5, 6, & 7 (where appropriate) and better discussed throughout the text. Specifically, what do uncertainties represent (e.g., propagated precision from fits, standard deviations, reported uncertainties from original geothermometers?), what confidence interval do these uncertainties represent, and what the uncertainties mean for the interpretations drawn from the data.

We have now included uncertainties in the text and corresponding error bars in the figures, and have also expressed what the uncertainties were derived from (e.g. Line 323). These are mostly based on standard deviations for the individual parameters, but for the discussion relating to the Schito and Corrado equation and the resulting conversion through Barker and Pawlewiscz, uncertainties are derived from the original equations.

3. More detail is needed on the organic matter comprising the samples and on the Raman analyses. Specifically:

a. What types of organic matter made up the extracted kerogen? Different organic matter types will have different Raman response due to differences in their molecular structures.

Organic matter types were not analysed, but we have highlighted the fact that changes in lithology may influence the resultant spectra. However, we are more interested in trends in the data and anomalies observed within the transects (which are now presented in more detail: Figure 6).

b. How does the kerogen isolation procedure change the Raman response? If this wasn't' tested, some text describing why this procedure isn't anticipated to impact the Raman spectra needs to be included.

This procedure has been performed on many examples where the carbon content may be low and shows no evidence of spectral shift:

J.S. Schmidt, R. Hinrichs, C.V. Araujo, 2017: Maturity estimation of phytoclasts in strew mounts by micro-Raman spectroscopy. International Journal of Coal Geology, Volume 173.

Muirhead, D. K., Parnell, J., Spinks, S. and Bowden, S. A., 2017: Characterization of organic matter in the Torridonian using Raman spectroscopy, Geol. Soc. London, Spec. Publ., 448(1), 71–80)

c. What mesh size were the samples powdered to prior to demineralization?

Samples were not powdered to a specific mesh size (though most were smooth to the touch) but were instead crushed for a length of time that was consistent between samples to minimise the chance of different durations potentially shearing the samples to varying extents, although the effect of this was thought to be minimal. See refs ready above

d. What grade of HCl and at what temperature was used in the demineralization procedure?

Room temperature and 10% HCl were used; these detailed are now included in the manuscript (Lines 236 and 239)

e. Were there entrained clay particles in the kerogen isolates that remained after demineralization? If so, could that impact your results?

During the demineralisation process, clay that was contained within the bulk sample tended to float to the surface of the solution, particularly during neutralisation/rinsing. Once dried, this clay could be scraped off the surface of the residue.

f. What evidence do you have that graphitization has not occurred in any of your samples?

We predict that due to burial depths of 6 to 8km it is extremely unlikely that graphitisation has occurred due to burial conditions. There were no intrusions to elevate temperatures and samples were taken away from the influence of hydrothermal fluids (with the exception of fault surfaces which could of course be conduits for fluids). Finally, none of the observed spectra were graphitic in nature.

g. How did you assess the presence/absence of thermal alteration from the excitation laser during Raman collection?

All acquisitions were the same length and duration, with laser power kept low. Samples were visually checked for signs of burning after each spectrum was recorded.

h. What microscope objective was used? What was the numerical aperture? What was the laser polarization?

The microscope objective was 50x magnification. Numerical aperture was 0.75 and laser was polarized from source.

i.      What function was used to smooth the data? How did smoothing the data impact the Raman parameters from the fits?

Smoothing the data aided the visual identification of spectral peaks. Where smoothed spectra still appeared to have significant noise, this was taken as an indication that errors due to the software fitting curves might be increased.

4. Figure 4 shows 'typical' changes in Raman parameters vs. temperature and strain. However, no discussion (or very little) is given to how these 'typical' trends were determined, especially for temperature. A better representation of this kind of analysis can be seen in Henry et al., 2019, Earth-Sci. Reviews. I recommend removing this figure (at least with regards to temperature) and directing readers to the trends shown Henry et al.

Figure 4 is intended to be a visual summary of general trends in Raman spectral parameters and is not to scale. Trends are based on data summarised in Henry et al. (2019) but are simplified further and are not tied to specific numerical values. The figure is simply to visually represent the written description in Section 4.3.

5. The discussion of the Raman band separation (RBS) parameter is unclear throughout and is incorect in several places. Specifically:

a. Section 5.2: "…RBS appears to show no consistent pattern throughout the stratigraphy…". To me, the RBS parameter shows a consistent, if weak, trend decreasing with depth.

We originally described this lack of trend because the variation between adjacent samples was often of the same or similar magnitude to the overall change through the stratigraphic section, so any apparent weak trend could feasibly be a result of this variation. However, we accept that it is still worth mentioning that there does appear to be a weak trend, as you point out. Therefore, we have included a note on this in Lines 348-349: "Raman band separation (RBS) varies through the stratigraphy (Fig. 5b), with what appears to be a prevalence of values >265 in the upper stratigraphy and <265 in the lower sequence."

b. Section 5.2: +/- 4 cm-1 for the RBS parameter is referred to as "high error". What does this error represent (see comment #3) and furthermore, this degree of uncertainty is <+/-2% the determined value. Why do the author's think this level of error is high?

The error is considered to be high because the variation between adjacent samples is of similar magnitude to the overall change observed through the whole sequence (see above comment). We have clarified this in Line 349: "The average error associated with each sample is +/- 4, suggesting that the change through the stratigraphic sequence is not significant."

c. Section 7.2: The statement "It is possible that the samples in this study are not of high enough maturity for a trend in RBS to be seen…." is not correct. Relatively low temperature organic matter show trends in RBS. In fact, both of the citations provided to

support this statement show RBS data with trends for low temperature source rocks. I do not understand where this statement comes from.

This statement originally stemmed from the high degree of variation in low-temperature RBS results presented in the studies cited; however, we accept that this statement is perhaps misleading. We have therefore adapted the manuscript to fit this: Lines 432-434 now read "Raman band separation (RBS) is reported to increase with increasing temperature (Fig. 4b; Zhou et al., 2014; Bonoldi et al., 2016; Sauerer et al., 2017), and so should increase with depth towards the basal thrust in our study."

d. Section 7.2: The statement "This suggests that frictional heating does not play a significant role in changing the Raman spectral parameters on thrust planes." is not supported by the data. RBS is calculated from D- and G-peak frequency. FWHM and peak height are other Raman parameters not included in the RBS parameter.

We suspect that our wording here was unclear and we thank you for highlighting this. In our discussion (Lines 434-440) we now include the following points: "If frictional heating on fault planes were the primary control on changes in RBS, and we assume an approximate instantaneous slip magnitude of ~1 m, then it would be expected that temperatures could rise by >400°C (Savage et al., 2014). This should be enough to produce a shift in RBS which is greater than the general variation we see in our samples. However, Nakamura et al. (2019) report that in addition to temperature, RBS is sensitive to lithology and the effects of fluids, which may explain the variable results we see in this study."

6. Line 372: "….we only observe a decrease in FWHM[d], whilst FWHM[g] changes very little…". This suggests that something weird is going on with either the samples or the analysis as this observation is fairly unexpected. More discussion is needed to explore this observation as FWHM[g] is usually considered one of the "best" indicators of maturity in Raman geothermometry.

We are aware of this discrepancy but thank you for pointing it out! It is beyond the scope of this study to investigate this as we are merely aiming to present the data observed in relation to how the parameters change across locally strained rocks; however, it is worth further investigation, as you correctly suggest.

7. Section 7.8: The statements "From our observations…..equation is less effected by strained environments than…" and "The Kouketsu equation is also more suited to strained environments." illustrate a real disconnect in the author's perspective on Raman geothermometers and what is *actually* being measured when you probe organic matter with a Raman instrument. The Raman response from organic matter is dictated by the molecular structure of the organic matter ensemble in the probe volume of the Raman microscope. Raman geothermometers work because the thermal alteration of organic matter structure as it reacts toward a graphite endmember is deemed irreversible. The interesting question this study is trying to ask is "Does strain, independent of temperature, change organic matter structure such that these effects need to be accounted for when determining a temperature from Raman spectra of geologic organic matter?". That is, all Raman geothermometers should be affected by strained environments if strain is changing the molecular structure of the probed organic matter. Stating that one geothermometer is more appropriate than another for strained environs misses this point. I strongly suggest reworking the discussion and conclusion with perspective on this point.

We are particularly grateful for this comment as it highlights the key points which we have made clear and those which we have not. Your (correct) interpretation of the key question that this paper is trying to answer was useful for helping us to refine the focus of the manuscript. We also appreciate that all geothermometers will be affected by strain if our results are correct; however, the point we tried to make was that since certain spectral parameters are more affected by strain than others, a geothermometric equation which utilises those parameters which are *less* affected by strain (e.g. FWHM[d]) would logically be *more* applicable in strained environments, although of course this only further highlights the simplicity assumed by many existed Raman geothermometers!

We now only include the Schito and Corrado equation in our manuscript and only do so as a discussion point. We mention the sensitivity of I[d]/I[g] and look at this in the context of our results, if the equation were to be applied (Lines 505-514): "The most significant term in the equation is I[d]/I[g], and our data shows that I[d]/I[g] is strongly affected by strain-related spectral changes. It therefore follows that the equation should be sensitive to strain, but the fact that not all strained samples produce calculated temperature shifts of the same direction or magnitude suggests that the process is more complex than simply strain or temperature having an effect. Regardless of cause, however, an error in calculated temperature of ±10°C in a stratigraphic sequence with an overall temperature range of only 25°C highlights the importance of context when estimating temperatures using this method. For example, if using this temperature data to reconstruct a burial history, then a strained sample might be 'out' by over a kilometre, or it might give the correct value. It is therefore important that more work is done to calibrate Raman geothermometers in rocks which have undergone strain in natural environments."

8. Finally, some discussion should be given to the work examining the Raman response of kerogen to high pressures in laboratory settings. Certainly, the rate of strain between natural samples (as examined here) and laboratory strained samples will be different, but I believe that prior laboratory experiments can provide much insight into the processes under study here. Start with:

Huang et al., 2010, In situ Raman spectroscopy on kerogen at high temperatures and high pressures, Phys. Chem. Minerals, 37, 593-600.

Thank you for this suggestion. We have added as part of Section 6.2 discussing RBS that "pressure also affects peak positions" in Line 441, and have found the references within the Huang paper particularly useful.

**Minor Revisions**

1. Line 30: define "reliable". This word is subjective without context.

The word "reliable" has been removed, with the sentence now reading "…to develop temperature equations that are based on Raman spectral parameters and are applicable across a range of settings…" (Line 27)

2. Line 40: The use of the term "organic carbon nanostructure" is misleading. Raman spectroscopy of complex geologic organic matter typically only reports on the aromatic character of the organics due to resonance effects (i.e., the electronic bandgap of the aromatic moieties matches the energy of visible wavelengths

commonly used as Raman excitation sources which pumps the response from these functional groups). Suggest rewording as "organic matter aromaticity".

Thank you for this suggestion. However, we have decided to continue with the term "nanostructure" because the aim of this paper is not to analyse the precise structure, bonds, or resonance of the molecules involved but rather to address changes in Raman spectral parameters due to strain. We do not try to discern what these changes precisely imply in terms of the molecular structure of the carbon involved as this is beyond the scope of this paper (but would be worth future study). The term "nanostructure" is only mentioned for the purpose of putting the spectral changes into some context; we do not wish to overcomplicate the concepts involved. We refer the interested reader to the appropriate literature (Line 2).

3. Line 44: Define "FWHM" at first usage.

Definition now included (Line 46). Thank you for pointing this out.

4. Line 154: Define "BRGM" at first usage.

Clarified as the French geological survey (Line 201).

5. Line 171: The statement "Raman spectroscopy measures the wavelengths of backscattered…." is not entirely true. First, Raman scatter occurs in all directions to a degree, not just in the backward direction. Second, Raman is used to measure much more than just different forms of organic carbon. Rephrase.

Thank you for pointing out this oversimplification. We have rephrased the sentence to read "Raman spectroscopy measures the wavelengths of radiation produced by inelastic (Raman) scattering during the de-excitation of electrons in different molecular bonds, in this case focussing on those involved in different forms of organic carbon." (Lines 222-224)

6. Line 172: More than just "peak temperature and strain conditions" are important for determining the molecular structure of geologic organic carbon. I would argue that biologic origin, depositional conditions, erosion, exposure to oxidants, and microbial activity are just as important as temperature and strain (and perhaps more so!).

This is a valid point and certainly highlights the depth to which organic carbon has been (and still needs to be) studied. However, due to the risk of overcomplicating what actually our methodology section, we have only included an additional note on biological origins here. The sentence now reads: "…depending on many factors during both deposition and burial: these include, but are not limited to, initial kerogen type, peak temperature, and the strain conditions…" (Line 225)

7. Line 173: The statement "Initially, the carbon will exist in the form of fossilized organic matter." is not correct. Initially all of this organic carbon was from living carbon-based life.

We have altered the statement so that it now reads: "In the initial stages of burial, the carbon will have a nanostructure resembling that of kerogen…" (Line 230)

8. Line 180: What does "excess inorganic carbon" mean? Excess of what?

The word "excess" is indeed unnecessary and has been deleted.

9. Line 185: Change "lots" to "co-adds".

Done, thank you.

10. Line 188: In figure 3 the caption states that a "Gaussian-Lorentzian hybrid" was used to fit the spectra. This is commonly termed a Voigt or pseudo-Voigt profile. Regardless, which profile shape was used to fit the data.

Thank you for pointing this out; we have changed the terminology to fit.

11. Line 204: The statement "The intensity of a single peak is a direct product of signal strength,…" is tautological. Raman intensity is proportional to the fluence of the input excitation source, the number density of oscillators in the probe volume, and the Raman cross-section (itself a function of the molecular polarizability).

Although of course correct, we consider this explanation to be too complex for the point we are trying to make. When dealing with sediments such as those in this study, absolute intensity of the spectral peaks can vary by an order of magnitude whilst intensity *ratios* remain constant, which is why intensity values of individual peaks are not used. We have, however, rephrased the sentence to make it clearer (Lines 269-274): "The intensity of a single peak is a direct product of signal strength, i.e., how many Raman-scattered photons come into contact with the detector. This can be affected by several factors including the amount of carbon present within the laser spot, or the strength of the laser. It is therefore more common to use the ratio between the D- and G-peaks ($I_d/I_g$), which will be characteristic of the nanostructural features regardless of signal strength."

12. Line 206: The statement "The G-peak is in fact a composite of three spectral bands…" is not correct. For less ordered carbonaceous materials the G-peak is best represented by a single peak, for higher metamorphic grade organic matter there can be another peak or two in there, but for graphene (arguably the most ordered carbonaceous material) there is only one G peak whereas for single-walled carbon nanotubes the G-peak is split into G- and G+ peaks. What I am saying is that Raman spectra of carbonaceous materials is incredibly complicated and so definitive statements such as this are inappropriate.

You are correct to highlight that such a definitive statement is inappropriate; the wording has been changed to reflect this (Lines 279-282): "The G-peak defined here can be considered a composite of up to three spectral bands (D2, G, and D3) depending on metamorphic grade, but at low maturities such as those in this study they are difficult to distinguish and can be collectively referred to as a single peak (Beyssac et al., 2002; Muirhead et al., 2021)."

13. Lines 216-217: What do you mean by "pure graphite"?

Changed to read "complete graphitisation" (Line 289)

14. Line 219: The statement "…shows an increase in RBS with increasing temperature at higher maturities…" is tautological. Increased temperature = higher maturity.

What you refer to as a tautology is intended to highlight that at lower maturities this trend is not always observed.

15. Line 245: The wrong paper by Barker and Pawlewiscz is cited. You are looking for Barker and Pawlewiscz, 1986, The correlation of vitrinite reflectance with maximum temperature in humic organic matter, Lecture notes in Earth Science, Vol. 5, Paleogeothermics, Edited by G. Buntebarth and L. Stegena, Springer-Verlag, Berlin.

Thank you for pointing out this mistake! This has since been corrected (Line 488).

16. Line 283: Quantify "weak trend".

Actual values have now been included in the text to quantify this (Line 358).

17. Lines 309-310: Remove "(although reversed)".

Changed wording for clarity, but we believe it is beneficial to leave it in (Line 442)

18. Line 373: The statement "…is kerogen-like carbon rather than coal,…" is misleading. Kerogen is operationally defined as insoluble sedimentary organic matter. Hence, coal is kerogen. Usually, coal kerogen is termed Type I, or gas-prone kerogen. Change statement accordingly.

We have changed "kerogen-like" to "amorphous" (Line 449.

19. Line 404: "…(as noted previously by Muirhead et al., in review)…". I don't love citing unpublished work. Also, this is definitely not "noted previously" as it is unpublished. Finally, this citation is not included in the bibliography. I suggest removing this citation.

This paper has now been published and hence the citation has been adjusted to fit.

20. Line 427: The statement "Since the Lahfid et al….." is unclear. Rephrase.

We have removed this section from the paper: see earlier comments.

21. Line 536: Space needed between "Michael" and "Raman".

Thank you for highlighting this.

---

## Author Comment (AC2)

Dear Dr. Rahl,

Thank you very much for taking the time to review our manuscript and for your highly constructive feedback – your comments have been extremely useful in helping to refocus our paper. As outlined in our response to A. Jubb, we have made the following significant changes:

- We have focussed the paper on the changes to individual Raman spectral parameters rather than geothermometric equations, only including the geothermometry as a minor discussion point later on in the text. We believe that this makes the manuscript far clearer and more focussed, not to mention easier for the reader to surmise the key points of the study.
- Transects across faults and shear zones are presented in more detail.
- Error ranges have been added to both text and figures.
- Methodology and definitions have been significantly tightened.

As per your suggestion, we have introduced a new figure which includes detailed transects of each locally strained locality, showing the precise changes exhibited by each spectral parameter (Figure 6). We have ultimately focussed the results and discussion around this figure, which we believe has brought greater clarity and focus to the manuscript. Having included this extra detail, we made the decision to omit the section dedicated to various geothermometers, as it was evident that this was confusing to the reader and that this section may be worthy of a study in itself. A future paper which dealt with the geothermometry section alone would be capable of doing greater justice to this complex subject, the scope of which would include all the different methodologies associated with specific geothermometers. In the meantime, the revised manuscript focusses in greater detail on the fine-scale transects, as you suggest, with a discursive note on the Schito and Corrado equation as an implication of our findings rather than an induvial set of results. By centring our results around the individual transects, we have additionally been able to discuss potential strain gradients. These strain gradients, and our definitions of 'strained' vs. 'background', are now explained in greater detail in lines 99-124, where we have included additional information. Please note that due to the different lithologies involved in each transect, it was more difficult to quantify the relative strain than in (for example) Kedar et al (2020) where the entire shear zone was contained within one lithology.

Thank you also for your specific comments, which have been addressed individually below. We have added error bars to our figures and specified them appropriately in the text, and have investigated where there are inconsistencies such as those highlighted in your comment above. Additionally, we have emphasised our discussion on other factors which may influence the changes observed in Raman spectra, also as per your suggestions above. Once again, we are grateful for your feedback and believe that this has greatly helped us to improve the manuscript.

Yours sincerely,

L. Kedar, C. E. Bond, and D. Muirhead.

**Specific comments**

Line 89 – "background" strain? Given the purpose of this study, assessing and describing this "background" strain is important. Can the strain in these rocks be quantified?

We agree that this could be better defined, and have therefore added a further few paragraphs (Lines 91-111) to explain the categorisation in more detail. It was harder to quantify the strain levels in the rocks than, for example, in our 2020 paper (Carbon ordering in an aseismic shear zone: implications for Raman geothermometry and strain tracking, Kedar et al., 2020), where the shear zone in question occupied a single lithological unit. In the current manuscript, several faults and shear zones involve multiple lithologies, and therefore most sampling was based on judgement in the field. This strategy is better explained in the additional text mentioned above.

Line 95 – one sample or a cluster? I don't understand what was done in practice here. How closely spaced were samples in a "cluster"? How were they averaged together? (Physically, or were analytical results averaged?)

If multiple samples were collected at an outcrop and these showed similar analytical results, these results were averaged.

Line 190 – I am confused on the analytical procedure used here. Above (line 185) it says that for each sample, 10 grains were studied, and that each grain was scanned three times. That would make 30 analyses per sample. Yet here it says "this process was carried out 3 times for each acquired spectrum, resulting in 30 analyses per sample" – This seems to be referring to the data processing steps, though I don't understand why performing these steps multiple times wouldn't produce the same results for each spectrum. And further, I don't understand where the 30 analyses stated here comes from (10 grains * 3 acquisitions per grain * 3 "processes carried out for each spectra" = 90, not 30).

Thank you for highlighting the confusion here. The 3x5sec acquisitions per grain produces one spectrum, which is then deconvolved 3 times. This entire process is carried out for 10 grains per sample. Therefore, 10 grains * 1 spectrum * 3 deconvolutions = 30 results. We have tried to clarify the fact that only 1 spectrum arises from the 3x 5-second acquisitions by adapting the sentence to read, "Each run comprised three co-adds of 5 second acquisitions to produce a single spectrum for analysis. This process was carried out on 10 individual grains from each sample." We have also removed the word "acquired" from line 214 to prevent further confusion.

As for the point about 3 deconvolution repeats, this is to minimise error from the user-guided part of the baseline subtraction process. The user must select pinning points along the spectrum to which the software fits a cubic spline interpolation. Particularly for poorer spectra, the position of the fitted/subtracted baseline can vary significantly depending on the exact pinning point selected. Of course, we could have taken this a step further and involved multiple users in order to minimise bias, but that is perhaps beyond the scope of this study. We have added a note about this in Line… reading "This process was carried out 3 times for each spectrum to minimise the error involved in the user-guided baseline removal process, resulting in 30 analyses per sample."

Line 245 – how is Ro_eq determined from the Raman parameters? (i.e., what equation is used?) It is strange to provide these equations here but still require the reader to go back to Schito and Corrado to see how the Raman parameters come into play. I think

there is no reason to include the equation for T1 here without also presenting the method to determine Ro_eq.

Thank you for highlighting this. We have now included the Schito & Corrado (2018) equation.

Line 281 - "high error causes these ranges overlap" What are the errors? How are these determined? They are not shown on the figure nor in the supplemental data table. Why not plot the error bars associated with each sample on the plots?

We have addressed this point by adding error bars to our figures.

Line 395 – "… or that the equation is not applicable in this instance due to the Raman parameters used"? What is meant by "due to the Raman parameters used"?

This sentence was originally rephrased to read, "This could be due to a weak geothermal gradient, or that the equation contains terms which are affected by strain, therefore altering the results." However, we have since omitted this section for clarity (see earlier comments).

Line 400 – this is difficult for the reader to evaluate because the equation incorporating this parameter is not provided

Addressed by adding the equation – see above response

Line 420 – "significant variation on a sub-km scale" – what, specifically, is meant here? (put a number of the variation; this is relevant for the following discussion)

Thank you for pointing out this imprecision. Our intention was to point out that values exhibited by adjacent samples varied to the extent that a trend was only visible on scales greater than a km, but you are right to point out that a number should be defined with such a statement. However, we have since omitted the section comparing the various geothermometers (see above general comment).

Section 7.8 – The RSCM thermometers investigated in this study are assessed here, based on how much the estimated temperatures differ for the "strained" and "background" samples. For instance, the Kouketsu thermometer is stated to be "more suited to strained environments" because the strained and background samples give similar results (line 431). But I think this misses the point. The goal of this entire process is to, as accurately as possible, estimate the peak temperatures experienced by rocks. A given thermometer equation can yield results that are insensitive to a particular factor (such as strain), but that does not mean that it produces reliable estimates. A key question is, does strain organize organic material in a fashion similar to increasing temperature? If so, I think the goal should be to build a model that connects the nature of the CM (as measured via Raman) to both temperature and strain. It would be ideal if the study had been conducted in a setting where independent estimates of temperature were available (like in the various studies used to calibrate versions of the thermometer). Without this, the authors need to rely on the internal consistency of the results, which can provide interesting insights but can't address the question of which of the applied thermometers gives the most reliable results in the study area.

We appreciate this well thought out comment and agree that an ideal study comparing various geothermometers would indeed include independent temperature estimates. Since we have decided to focus the revised manuscript on the individual parameters only, with greater emphasis on the strained transects, we have omitted the section which compares the various geothermometers (apart from a discursive point on applying our results to the Schito and Corrado equation). This is beneficial both in terms of clarifying the aims of the current paper but also in that it opens up the opportunity to expand the scope of a future study which deals with the comparison of these geothermometers as its principal aim. Such a study would of course include independent temperature estimates which we were unable to obtain for this study due to time, funding, and travel constraints.

Line 432 – "lower than expected for the region…" – on what is this expectation based? In the earlier part of the paper, all that is said about these rocks is that they are "subgreenschist facies" – there don't seem to be any detailed controls on peak temperature here.

Due to various limitations (see above comment) we were unable to obtain independent temperature constraints beyond those gained through simple calculations relating to burial depth and metamorphic facies based on the exiting literature. As it happens, the section referred to in this specific comment has now been omitted (see above).

Line 433: "shows variation in temperature predictions on a sub-km scale, making it less suitable for general use" – I'm not sure what is meant by "general use" here, and I also don't follow the logic. I interpret this to mean that the temperature estimates made using this thermometer are noisy, and therefore unreliable. I worry the authors underestimate the uncertainty inherent in these techniques.

Thank you for pointing out another instance of imprecise language in the manuscript; this is something that we have endeavoured to rectify throughout the revised text. Your interpretation here is correct; however, we are also aware of the "inherent uncertainty" that you mention here. Once again, this geothermometer comparison section has been omitted for the purpose of clarifying the focus of the paper.

Line 435: "… demonstrates a more consistent error…" – what is a "more consistent error"? There are no independent estimates of the temperature for these rocks, so there doesn't seem to be a way to estimate the accuracy of the various thermometers. Do the authors mean that the results for the Lahfid equation for the strained samples are consistently biased in the same direction? I think that is different than having a "consistent error", which I would take to reflect the magnitude of a temperature difference from a true value.

Once again, thank you for highlighting our choice of language – and again, your interpretation is correct. We refer to a consistent shift direction, and although the Lahfid equation is no longer used, we have ensured that this terminology is used in the rest of the paper (e.g. Line 369).

Line 435: "…the predicted temperatures are more in line with those predicted for the area…"  Again, based on what? There don't seem to be any independent temperature estimates.

This is based on burial depth – however, we no longer compare multiple geothermometers in this work (see earlier comments).

Line 460: "… an equation is required that can resolve temperature changes over hundreds of metres at the least" – is this even feasible? There are errors in all of these measurements.

This is an excellent question, and not one we can even attempt to answer within the scope of this study! However, future work and the continued development of Raman geothermometric techniques will hopefully provide an answer.

Line 461: "we conclude… choosing the most appropriate equation is complex and dependent on multiple factors" – what are the "multiple factors"? The preceding discussion suggests that strain is one, but I'm not sure what else is meant here.

To clarify this, we now refer only to strain in the concluding remarks, and as mentioned above no longer compare multiple equations. We do however mention that "the fact that not all strained samples produce calculated temperature shifts of the same direction or magnitude suggests that the process is more complex than simply strain or temperature having an effect" (Line 507)

Line 465: "The use of multiple parameters… suggests that the equation should be relatively insensitive to strain" – I don't understand this. If any of the parameters that go into the temperature estimate are influenced by strain, why wouldn't that influence the results? Why would including other parameters limit the influence of parameters sensitive to strain?

The reasoning behind this statement follows the idea that an equation which is based on one parameter only will be 100% affected by that parameter's sensitivity to strain. If multiple parameters are used in an equation, and not all of them are altered by the introduction of strain, then the result of the equation will not still be affected but not to the same degree.

Line 468: "The use of multiple terms in the equation may help to produce more reliable results (as the influence of different parameters interact)" – why? The logic here is not clear to me. I think what matters is the inclusion and appropriate weighting of the relevant parameters, not the number of terms in the equation.

Our point here refers to the weighting of parameters, as you suggest, but our wording was not clear. As we no longer compare multiple geothermometers, we do not discuss the relative benefits of weighting certain parameters, only alluding to this point in Line 505 where we state "The most significant term in the equation is I[d]/I[g], and our data shows that I[d]/I[g] is strongly affected by strain-related spectral changes. It therefore follows that the equation should be sensitive to strain…"

**Technical corrections**

Line 21 – should be "up to 140"

Since we no longer analyse multiple geothermometers we have omitted this statement from the abstract.

Line 100 – grammar (replace semi-colon with " and…")

Changed – thank you.

Line 171 – Reword; the text here implies that Raman spectroscopy works only on organic carbon.

Text now reads "Raman spectroscopy measures the wavelengths of radiation produced by inelastic (Raman) scattering during the de-excitation of electrons in different molecular bonds, in this case focussing on those involved in different forms of organic carbon." (Line 222).

Line 178 – Is "pertaining" the right word here?

Perhaps not – this has been changed to "approaching" (Line 233)

Line 261 – should be "detail", not "detailed"

Changed – thank you.

Line 277 – units should be included with these values

We appreciate that unitless values look strange, but as we are presenting ratios we cannot give them units.

Figure 4 – why are some of the grey lines dashed rather than solid? There is no explanation for this on the figure or in the caption. Other than this and the lack on uncertainties shown on the data, the figures are very attractive and quite helpful, especially the maps and cross sections (good work!).

Thank you for pointing out our failure to explain this. The dashed lines are to indicate where trends are not directly indicated in the literature but are inferred from related parameters.

---

## Author Comment (AC3)

Dear Dr. Nakamura,

Thank you for taking the time to review our manuscript and construct such useful comments. We have found your points very helpful in structuring a revised manuscript, which now focusses on highlighting the specific changes in Raman spectral parameters in detailed transects across the strained localities and comparing these to the burial trends, rather than attempting to compare various geothermometric equations. We believe that this makes the point of the manuscript clearer, whilst also allowing for a future follow-on paper to deal with the geothermometers individually and therefore expanding the scope to include methodological detail appropriate for each equation, as well as independent measures of temperature. We believe that this approach has greatly benefitted the manuscript as we can now focus on the original aim of the study, which was to assess the ways in which strain affects Raman parameters.

As outlined in our responses to other reviewers, our major changes are as follows:

- We have focussed the paper on the changes to individual Raman spectral parameters rather than geothermometric equations, only including the geothermometry as a minor discussion point later on in the text. We believe that this makes the manuscript far clearer and more focussed, not to mention easier for the reader to surmise the key points of the study.
- Transects across faults and shear zones are presented in more detail.
- Error ranges have been added to both text and figures.
- Methodology and definitions have been significantly tightened.

Please see below each of the major and minor comments for specific responses and changes made. We thank you again for your time and effort in helping us to refine the manuscript. Our comments are highlighted in blue text.

Yours sincerely,

L. Kedar, C. E. Bond, and D. Muirhead.

Major comments

1. Before you start to discuss the effect of strain in natural deformed rock samples, you should state a more detailed discussion on the peak metamorphic conditions of "background" samples under low-grade metamorphism. In your manuscript, there are no comments on the peak temperature conditions for "background" samples based on different thermal indicators (such as vitrinite reflectance, mineral assemblages of mafic rock, and illite/chlorite crystallinity) and previous literature. Hence, I am not sure which thermometry is more suitable for peak temperature estimation. Although carbonization depends sensitively on other effects such as tectonic deformation, fluid activity, lithostatic pressure, and duration of heating, there is no doubt that peak temperature is the most important factor during carbonization/graphitization. Therefore, you should compare between estimated temperatures by each thermometry and individual thermal indicators such as IC or mineral assemblages before discussing the effect of strain. In particular, the difference in peak temperature conditions inferred from three different thermometry is very interesting. Each thermometry was empirically calibrated by other

temperature estimations at different localities. This difference might be the key to understanding the effect of other factors including the tectonic deformation during carbonization.

Thank you for this thoughtful comment. We accept that in an ideal world, there would have been at least one method used to independently estimate peak temperature. However, as a result of time, funding, and travel restrictions, we were unable to carry out any of the additional methods listed above. This is part of our reasoning for omitting the geothermometer comparison from the revised manuscript, as we believe that putting it into a dedicated paper in future which deals with these independent temperature measurements would do it greater justice. We do, however, refer to estimated peak temperatures calculated using burial depths obtained from previous literature (Lines 379-382)

2. Three thermometers you applied are optimized for "dispersed" organic material in pelitic rocks, not marl or carbonate rocks. In general, the chemical structure of organic material (i.e., type II kerogen) in carbonate rocks is largely different from that of organic material (i.e., type III kerogen) derived from terrigenous sediments such as pelitic or psammitic rocks. Therefore, you should discuss the effect of precursor material between carbonate rocks and pelitic rocks before discussing the effect of strain.

Please see the previous comment: such precise detail could be easily discussed in a future dedicated paper dealing with the geothermometers as a focus to the study. Our main aim of the revised manuscript is to focus on the ways in which individual Raman spectral parameters deviate from any observed burial trend in locally strained samples, and how these changes are distributed throughout fine-scale transects across zones of high strain.

3. If the estimated temperatures reflected true burial temperatures, the peak temperatures inferred from Lahfid thermometry are inappropriate because it was calibrated in the range between ~200 and 320 degrees C (see a calibration line in figure 4 by Lahfid et al. 2010). Most estimated temperatures (~100-200 degrees C) you demonstrated in Figure 7 are out of calibration range. In addition, I am not sure why you did not apply for Rahl's thermometry in your study area. Please describe the reason why you select three thermometry.

The original reason for comparing the various geothermometers was to see how they were affected in zones of high strain, rather than being too concerned with the exact temperature values produced by the equations over the complete sequence. However, we appreciate that this could introduce more uncertainty into the results. We therefore think that our revised manuscript – in which we focus mainly on changes in the individual parameters rather than comparing geothermometers – is a more meaningful way to present our results.

4. In L213-215: It should be borne in mind that the significant decrease in ID/IG ratio by Nakamura et al. (2015) and Kirilova et al. (2018) was observed under brittle deformation of "fully ordered" graphite, not amorphous carbon and coal. Other literature mainly treated amorphous carbon or coal as a starting material to assess the change in ID/IG ratio during deformation. The crystallinity of starting material is completely different. Therefore, I think it is problematic to compare between the spectral evolution of graphite by deformation and that of coal and organic material.

It was never our intention to directly compare these different starting materials to one another as we realise that they are indeed completely different, so we are grateful that you have pointed out the fact that this is how it comes across. We have endeavoured to clarify that these situations are incomparable in the figure caption for Figure 4, where we believe the confusion has arisen. This is why Part (a) of Figure 4 has two trend lines: one for amorphous carbon and one for a more crystalline starting material.

5. Raman measurement using a powder sample is a good method to avoid polishing damage during making thin sections. However, it is difficult to recognize the laser damage on the OM surface. According to Nakamura et al. (2019), irradiation-induced depressions of 1.02-3.71 μm are observed at powers of > 0.7mW. Your measurement settings (<3mW) during micro-Raman spectroscopy are slightly higher than the threshold of laser irradiation we found (~1mW). In general, the registrability of laser irradiation is dependent on the crystallinity of OM and laser wavelength. Your samples are much weaker than the natural OM we analyzed. Hence, it is necessary for more careful analysis to avoid laser-induced heating and amorphization during measurements. It should keep in mind that down-shifting of D and G bands occur easily by laser-heating.

Thank you for pointing this out – we made an error in the writing of the methodology and actually used only 10% laser power which equated to 0.3mW at the sample surface. Therefore, burning was not an issue. We have corrected the text to reflect this (Line 241).

6. L35: Before you comment on the fold-thrust systems, it is better to discuss on the "ductile deformation" may enhance recrystallization of natural OM (See Ross and Bustin, 1990 and Bustin et al. 1995).

Thank you for highlighting this; we believe we discuss ductile deformation in sufficient detail in our description of the outcrops (e.g. Lines 206 to 219). See also Kedar et al. (2020).

Sincerely,

Yoshihiro Nakamura

Geological Survey of Japan, AIST

Specific comments

L13: I am not sure which parameter you indicate. Please indicate the four most common Raman spectral parameters and ratios in this sentence.

These have now been added (Lines 11-12): "…the most common Raman spectral parameters (peak width, Raman band separation) and ratios (intensity and area) change…"

L15: Please specify the D, G band, IG/IG, FWHM [d], Raman band Separation, and R2 ratio.

*See above comment.*

L21: upto => up to

*Thank you for pointing out this mistake*

L27: Ferrari and Robertson (2001) is inappropriate for reference. This paper discussed on wavelength dependence of Raman spectra of amorphous carbon under excitation from NIR to UV lasers.

*Thank you for highlighting this oversight. The reference has been removed from this sentence.*

L43: $I_D/I_G$ , I[D]/I[G] or R1 ratio are more commonly used.

*Thank you. We have chosen the notation to be consistent with Kedar et al. (2020).*

L44: peak area ratio (R2) => peak area ratio [R2 ratio = D1 band / (D1 + D2 + G bands)]

*As the R2 ratio is defined in more detail later, we felt that to define it to this extent here would read clumsily.*

L84: It is difficult to recognize the "level of strain" in natural rocks. Have you ever found more quantitative strain markers in your rock samples (grain size distribution of recrystallized quartz or strain marker such as Radiolaria)? Your classification "background" and "strained" is hard to understand for readers including referees. Please revise to be clearer.

*This was also highlighted by J. Rahl in his comments – we have added a more detailed description as to how background vs. strained samples were defined (lines 99-124). It was harder to quantify the strain levels in the rocks than, for example, in our 2020 paper (Carbon ordering in an aseismic shear zone: implications for Raman geothermometry and strain tracking, Kedar et al., 2020), where the shear zone in question occupied a single lithological unit. In the current manuscript, several faults and shear zones involve multiple lithologies, and therefore most sampling was based on judgement in the field. This strategy is better explained in the additional text mentioned above.*

L92-93; If so, it is better to describe the detailed occurrences of dispersed organic materials in background and strained rock samples.

*Thank you for this suggestion. A comment has been added to Line 133 which reads, "most organic material was located between calcite grains and within seams of insoluble material."*

L161-164: Effect of ductile and brittle deformation in natural rock samples is very important to assess the strain-induced carbonization or amorphization of natural OM. Please state more detailed comments how ductile and brittle deformation took place.

*It is beyond the scope of this study to fully analyse how the deformation took place on multiple scales and is only mentioned in Section 3.2 to provide some context when describing the samples analysed.*

L186-192: Peak deconvolution and fitting function are strictly defined by each equation. For Kouketsu equation, the peak temperature (= D1 band FWHM) fluctuates drastically depending on whether the D4 band at 1250 $cm^{-1}$ is fixed or not. Hence, it is very important to follow the recommended method.

Thank you for highlighting this. Now that our paper focusses on the individual parameters changing with strain rather than comparing different geothermometers, it is worth bearing this point in mind for any future paper which might focus on the thermometers themselves.

L200: $cm^{-1}$ = > $cm^{-1}$

Well noticed!

L209: kerogen-like carbon is not commonly used. Amorphous carbon? In addition, Beyssac et al. never comments on carbonization in this paper. It is better to refer to other papers such as Levine, (1993) and Oberlin, Bonnamy, & Rouxhet, (1999). See Levine, J.R., (1993). Coalification: The Evolution of Coal as Source Rock and Reservoir Rock for Oil and Gas. In Law, B.E., & Rice, D.D., (Eds.), Hydrocarbon from Coal, 38 (pp. 39–77) and Oberlin, A., Bonnamy, S., & Rouxhet, P.G., (1999). Colloidal and supermolecular aspect of carbon. In Thrower P.A., & Radovic L.R., (Eds.), Chemistry and Physics of Carbon, 26 (pp. 1–148). New York: Marcel Dekker, Inc.

Thank you for these suggestions and advice. "Kerogen-like" has been changed to "amorphous" (Line 279), and we have followed up on the correct citations.

L250-253: This is not the equation I know! Please explain why you change the parameter from RA1 to R2 ratio.

The two ratios – although not exactly the same – are similar in that they both deal with peak areas. The intent was to investigate how the thermometer results would change with strain and so exact results were not considered too important. However, the Lahfid equation is now not included (see above comments).

L269-275: Please add one sigma error bar in your figures. Your figures are unkind to readers and referees when I check the variation of Raman data between background and strained rock samples. If possible, please add the table in your manuscript.

Error bars have been added to figures where appropriate.

L329: The change in ID/IG ratio is much complex process. According to comparison between natural Organic material and pyrolysis samples by Nakamura et al. (2019), the $I_D/I_G$ ratio shows both trends in increase and decrease with increasing peak temperatures (Figure 5d). On the other hand, simple pyrolysis experiment suggests that the OM shows a systematic increase in the ID/IG ratio with increasing pyrolysis temperature and heat treatment time (Figure 7d).

We are aware that ID/IG changes are indeed complex and likely to be impacted by multiple factors, as you suggest. There is a lifetime of study in this field! We have added the suggested work as a citation (Line 397).

L358-359: I don't think so. Please refer to Nakamura et al. (2019). The significant change in RBS is widely observed in the temperature range between 180 and 280 degrees C (See Figure 5e). The systematic change in RBS is consistent with the change in illite crystallinity.

Thank you for pointing out this oversight. We have changed the text accordingly (Line 432) and added details from Nakamura et al. (2019) (Lines 438-440).

L365: plane,. => plane.

Changed. Thank you.

L383: Please add the mean value and one sigma error of R2 ratio. I am not sure this drop (6-20%) is important or not.

Errors have been added.

L394-395: The equation by Schito and Corrad (2018) was calibrated in the range of %Ro between 0.3 and 1.0 % (L242-244). I think that applicable range of vitrinite reflectance is much lower temperatures than expected temperatures under sub-greenschist facies. Have you checked the vitrinite reflectance of studied samples before micro-Raman analysis?

This is perhaps correct, although calculated values were within the applicable range of temperatures. The intention of looking at this geothermometric equation is to investigate how the results might potentially be affected by strain in certain samples, and therefore how they deviate from surrounding values, rather than the precise numbers being important. We have changed the emphasis of the paper to focus more on the parameters anyway, and now only include the Schito and Corrado equation as a discussion point. As for vitrinite reflectance, we were unable to carry out this analysis for this particular study owing to travel restrictions and time constraints, but perhaps if the aforementioned follow-on study were conducted, this would form an integral part of it.

L401-402: See an above specific comment in L358-359.

See above response.

L406: +/-10°C => ± 10 °C

Format corrected.

---

## Author Response (AR2)

Dear Editors,

We would like to sincerely thank Aaron Jubb for taking the time to review our revised manuscript and for presenting his ideas for minor revisions. The effort is much appreciated. All suggested changes have been made and are highlighted below, and in the PDF containing tracked changes (included as comments).

Yours sincerely,

Lauren Kedar

Clare E. Bond

David K. Muirhead

1. Page 9, Line 228: "Gaussian curve fit". In Figure 3 caption this is called a "Gaussian-Lorentzian hybrid". Fix this inconsistency.

*This has been fixed.*

2. Figure 7: Add error bars to data points.

*Error bars have been added.*

3. Page 16, Lines 410-413: FWHM[g] decreases with temperature are observed for more than just coals. See Henry et al., 2019, Earth Science Reviews, Figure 5a. I would change sentence slightly to reflect this.

*Sentence now reads: "This is also observed in the case of other starting materials including kerogen (Henry et al., 2019). However, since kerogen (and organic carbon in general) can take on such a wide variety of forms, it is easily possible that a trend observed in one case may not be strong in another."*

4. Page 18, Line 445: Define %Ro at first usage.

*%Ro now defined.*

5. Page 18: I am curious how the authors arrived at a temperature error of +/-10C from the Barker and Pawlewiscz equation considering the large spread in that data set. I would have thought the uncertainty would be much higher, especially considering that the errors on the Raman parameters are going to propagate through the calculation. Informed readers may also have this curiosity so I would encourage the authors to add a sentence detailing how this estimation was made.

*Thank you for highlighting the lack of clarity in our discussion – we have changed the wording slightly to avoid the use of the term "error" because this is misleading, suggesting statistical calculation, when we are actually just referring to an observed shift from background levels in the strained samples. The sentence now reads: "Regardless of cause, however, if a strained sample can produce a difference in calculated temperature of 10°C in a stratigraphic sequence with an overall temperature range of only 25°C, then context is important when estimating temperatures using this method."*